# HMGA1 orchestrates chromatin compartmentalization and sequesters genes into 3D networks coordinating senescence heterogeneity

Ioana Olan [1,10], Masami Ando-Kuri[1,7,10], Aled J. Parry[1,8,10], Tetsuya Handa [1], Stefan Schoenfelder [2], Peter Fraser[3,9], Yasuyuki Ohkawa [4], Hiroshi Kimura [5], Masako Narita [1] & Masashi Narita [1,6] ✉

HMGA1 is an abundant non-histone chromatin protein that has been implicated in embryonic development, cancer, and cellular senescence, but its specific role remains elusive. Here, we combine functional genomics approaches with graph theory to investigate how HMGA1 genomic deposition controls high-order chromatin networks in an oncogene-induced senescence model. While the direct role of HMGA1 in gene activation has been described previously, we find little evidence to support this. Instead, we show that the heterogeneous linear distribution of HMGA1 drives a specific 3D chromatin organization. HMGA1-dense loci form highly interactive networks, similar to, but independent of, constitutive heterochromatic loci. This, coupled with the exclusion of HMGA1-poor chromatin regions, leads to coordinated gene regulation through the repositioning of genes. In the absence of HMGA1, the whole process is largely reversed, but many regulatory interactions also emerge, amplifying the inflammatory senescence-associated secretory phenotype. Such HMGA1-mediated fine-tuning of gene expression contributes to the heterogeneous nature of senescence at the single-cell level. A similar 'buffer' effect of HMGA1 on inflammatory signalling is also detected in lung cancer cells. Our study reveals a mechanism through which HMGA1 modulates chromatin compartmentalization and gene regulation in senescence and beyond.

Cellular senescence, a stable exit from the cell cycle in response to diverse stimuli, is accompanied by drastic shifts in cellular functionality constituting the SASP, typically represented by the expression of genes with high tissue specificity[1]. Consistently, senescence has been linked to epigenetic and transcriptional alterations[2]. In addition, high-throughput chromatin conformation (Hi-C) studies showed that senescence is associated with a unique 3D chromatin profile[3–8]. Based on Hi-C profiling, chromatin can be broadly divided into active (A) and repressive (B) compartments, which become more separated in senescence, marked by increased long-range B-B contacts and reduced A-B contacts[7]. This is consistent with the formation of senescence-associated heterochromatic foci[9,10] (SAHFs), which are best described in the context of oncogene-induced senescence (OIS).

High Mobility Group A1 (HMGA1) is highly expressed in embryonic tissues[11] and many cancer types[12]. It also plays a critical role in SAHF formation and the stability of senescence arrest[7,13,14]. Despite the

physiological and pathological relevance of HMGA1[11], its precise roles in coordinating genome organisation and gene expression are poorly understood. This is due to the lack of comprehensive understanding of its gene regulatory mechanisms. HMGA1 has three AT-hooks, which bind AT stretches of DNA. Consistent with its preference for genomic regions with a high AT content, often linked with heterochromatic genomic regions, HMGA1 can be detected microscopically in heterochromatin, including SAHFs, in which HMGA1 is an essential structural component[13]. Paradoxically, in contrast to this imaging data, HMGA1 has been described as a gene activator via direct binding to specific gene regulatory elements (promoters and enhancers), despite not possessing transcription factor activity per se[11].

Here, we devised a strategy to resolve the contradictory roles that have been attributed to HMGA1. We integrate bulk and single-cell transcriptomic data, HMGA1 chromatin occupancy and other epigenomic profiles, as well as Hi-C 3D genome networks to quantify the impact of HMGA1 loss on chromatin organization, and to assess its gene regulatory role in OIS. Our data support an alternative mode of gene regulation by HMGA1 achieved by the modulation of chromatin environments (Supplementary Fig. 1a) and representing an additional layer of gene regulation through global chromatin configuration, rather than the traditionally characterised local cis/trans impact. HMGA1 occupancy is pervasive and heterogeneous across the genome, and it drives chromatin compartmentalization, thus reinforcing both key gene activation and repression during senescence. Considering the oncofoetal nature of HMGA1, our findings may have broad implications.

## Results

### HMGA1 modulates cellular functionality by affecting both up- and down-regulated genes in OIS

To characterise HMGA1-dependent gene regulation, we utilized the oncogenic RAS-induced senescence (OIS) culture model, IMR90 human fibroblasts expressing a 4-hydroxytamoxifen (4OHT)-inducible ER:HRAS$^{G12V}$ fusion protein[15]. These cells were stably transduced with a well-characterised shRNA against HMGA1 (shA1) in a miR30 design or control miR30 backbone[13,16,17], and ER:HRAS$^{G12V}$ was then induced with 4OHT for 6 days to establish OIS or OIS-shA1 cells. As previously shown[13], HMGA1 depletion suppressed SAHF formation with little impact on senescence arrest, probed by cell cycle markers (Fig. 1a, Supplementary Fig. 1b). 967 and 1,365 genes were significantly differentially expressed in proliferating (Grow, no 4OHT) and OIS conditions, respectively, with shA1 (Supplementary Fig. 1c). Typical SASP components, such as inflammatory cytokines (e.g., IL1B, IL8, IL6, and CXCL1-3) and matrix-degrading enzymes (e.g., MMP1 and MMP3), were up-regulated in OIS cells, but they were further up-regulated in OIS-shA1 cells (Fig. 1b, Supplementary Fig. 1d). This suggests that HMGA1 acts as a transcriptional buffer for the SASP. We confirmed that the IL-8 and IL-6 proteins were indeed more abundant in OIS-shA1 than OIS cells (Fig. 1a). We also performed immunofluorescence experiments for IL-8 and HMGA1 and confirmed the up-regulation of IL-8 in OIS-shA1 compared to OIS (Fig. 1c, IF and Supplementary Fig. 1f). Similar effects were observed in the transcriptomic profiles of proliferating IMR90 cells with shA1, with substantial up-regulation of SASP genes (Supplementary Fig. 1c, e), suggesting that HMGA1 has a modulatory effect on the expression of these genes in IMR90 fibroblasts.

While the SASP can be seen as a gain-of-function phenotype of senescence, a loss-of-function phenotype is also well described: senescent fibroblasts tend to lose fibrogenic activities[1,18]. Interestingly, HMGA1 also modulated the expression of genes involved in fibrogenic activities, with HMGA1 depletion leading to further down-regulation of ECM components, e.g., CTGF, FBLN2, FBN2, and BGN (Fig. 1b). In contrast, the effect of HMGA1 on cell cycle genes was modest at the cell population level (Supplementary Fig. 1g), consistent with the phenotypic observation of limited effect of HMGA1 on the cell cycle arrest.

These data suggest that HMGA1 acts as a key modulator of the major functional shift associated with senescence, for both up- and down-regulated genes.

### HMGA1 binding is linked to gene repression and weak enhancers

HMGA1 has been proposed to act as an 'architectural transcription factor (TF)', facilitating the binding of other transcriptional activators by changing the structure of DNA[11], but the mechanism behind such activity is unclear. To address this question, we performed highly optimised HMGA1 ChIP-seq experiments in IMR90 cells (growing and OIS conditions) and generated high-resolution signal distributions. We observed almost 400,000 discrete peaks in each condition (Fig. 1d), demonstrating that HMGA1 binds the genome pervasively, compared to other structural proteins (e.g., ~40,000 CTCF peaks in IMR90 cells[8]). Similar results were obtained in H1299, a human non-small cell lung cancer (NSCLC) cell line that express endogenous HMGA1 (Fig. 1d). We further validated the HMGA1 antibody specificity for ChIP-seq in HMGA1-deficent H1299 cells that we generated using CRISPR/Cas9 gene editing (Supplementary Fig. 1h, i).

Consistent with the high affinity of HMGA1 for AT-rich regions due to its 3 AT-hook and with previous reports correlating HMGA1 binding to AT-content[19], HMGA1 peaks had high AT% (avg. 72%). Motif analysis revealed highly AT-rich motifs (Supplementary Fig. 1j), similar to motifs previously associated with HMGA1 and other proteins which preferentially bind AT-rich motifs, such as ARID3A[20]. HMGA1 binding (signal binned in 1 kb windows) correlated positively with AT% in the control cells (Fig. 1e), and even more in OIS (Pearson 0.42 in Grow and 0.47 in OIS). 95% of the 1 kb bins with over 70% AT content were bound by HMGA1.

HMGA1 binding in both the Grow and OIS conditions in IMR90 cells showed the formation of broad binding profiles with a high density of peaks (Fig. 1f). We defined 'HMGA1-dense' regions by determining the inflection point of the distribution of the number of peaks in overlapping bins (100 kb rolling windows with 5 kb steps) and stitching together regions with a peak density greater than this threshold. 'HMGA1-dense' regions were similar in shape to H3K9me3 peaks (a marker of constitutive heterochromatin), with an average size of 450 kb (Fig. 1f), consistent with HMGA1's localization at heterochromatin regions[13,21] based on microscopic imaging (Supplementary Fig. 2a). HMGA1-dense regions showed good overlap with Lamin B1-associated domains (LADs) and H3K9me3 peaks in proliferating IMR90 cells (Fig. 1g). The HMGA1-dense H3K9me3 peaks overlapping were wider, more AT-rich and contained fewer genes than other H3K9me3 peaks (Supplementary Fig. 2b). While these HMGA1-dense H3K9me3 peaks tended to have overall less Lamin B1 binding than other H3K9me3 peaks, they showed a more pronounced Lamin B1 loss during OIS (Supplementary Fig 2c). Considering the critical role for both Lamin B1-loss (a hallmark of senescence) and HMGA1 in SAHF formation[13,22]. HMGA1-dense H3K9me3 regions are likely to contribute to SAHFs.

We compared the HMGA1 signal to epigenomic features summarized at high- (1 kb bins) and low-resolution (200 kb bins). At low resolution, consistent with the visual inspection (Fig. 1f), HMGA1 was positively correlated with Lamin B1 and H3K9me3 signal and anti-correlated with euchromatic histone marks (Supplementary Fig. 2d, 200 kb bins). However, at high-resolution, HMGA1 binding was independent of most of these features (Supplementary Fig. 2d, 1 kb bins) and HMGA1 peaks often co-localized with a 'valley' in the signal of other histone marks (Supplementary Fig. 2e).

Notably, despite covering 30% of the genome, HMGA1-dense regions only accounted for 36% of all HMGA1 peaks and, at high-resolution, HMGA1 displayed a discrete, well-defined peak pattern (Fig. 1f, bottom), more similar to that of euchromatic marks or TF footprints, than to the broad profile of heterochromatic histone marks. Annotation of the HMGA1 peaks showed significant overlap with

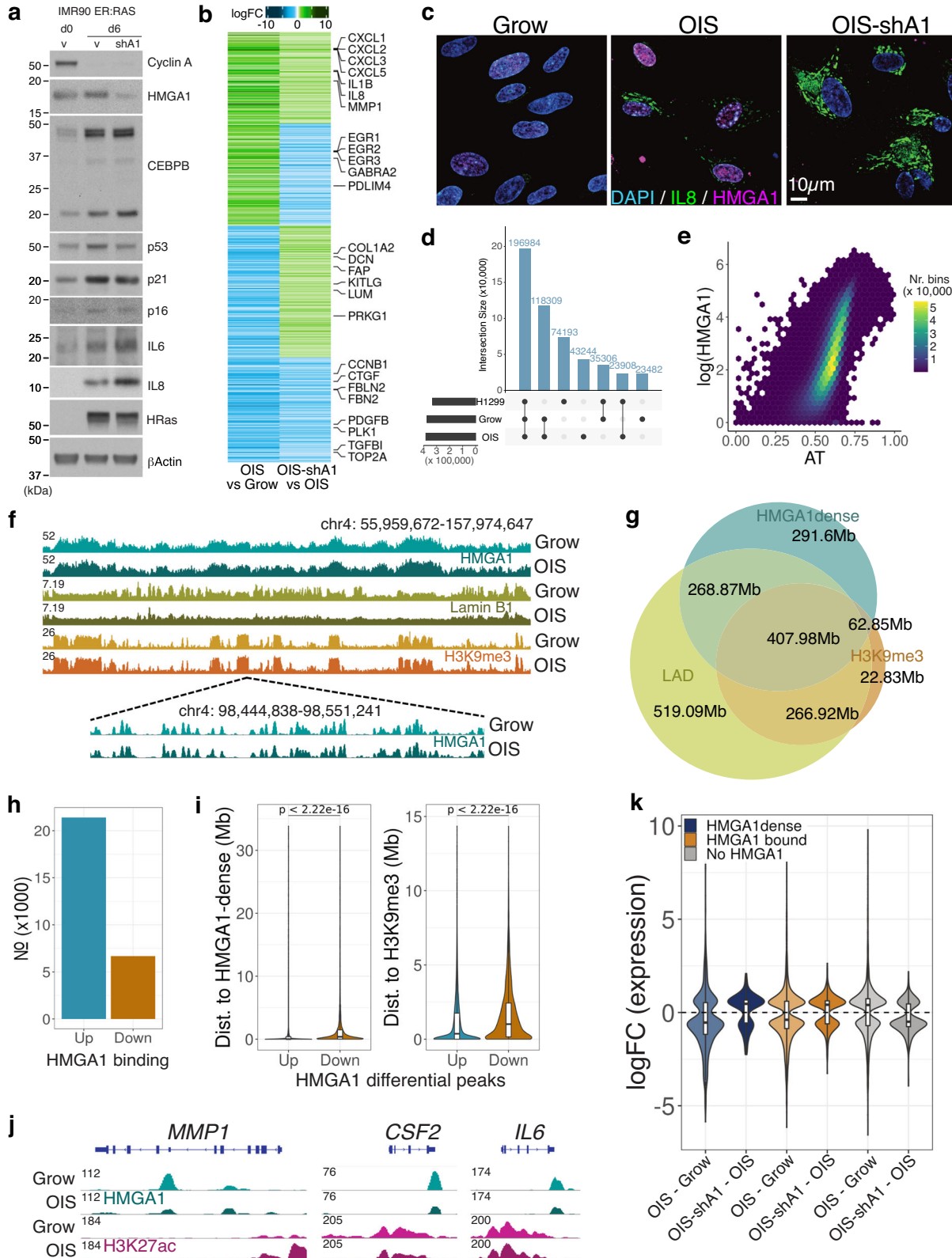

intergenic regions (59% of the peaks) (Supplementary Fig. 2f), likely due to the high density of HMGA1 peaks observed in gene-poor heterochromatic regions. However, a large proportion of peaks (40%) overlapped gene introns, which prompted us to repeat the analysis using only 'euchromatic' HMGA1 peaks (i.e., not overlapping H3K9me3). These peaks were enriched over genic elements: exons, introns, exon-intron junctions, and 3′UTR regions.

We found that very few promoters were bound by HMGA1 (Supplementary Fig. 2f, only 7% of all promoters, out of which only 2% were active), despite previous reports attributing promoter-binding roles to HMGA1-mediated gene activation[11] and the genome-wide abundance of HMGA1. This tendency may reflect the low enrichment of AT at promoter regions[23]. Genes with HMGA1-bound promoters mainly clustered in olfactory receptor (OR) family and type-I interferon loci

**Fig. 1 | The HMGA1 effect on transcription and genome-wide binding distribution in proliferating and OIS cells. a** Protein-level changes of key senescence genes in proliferating IMR90 (Grow; day 0), OIS (day 6) with and without shHMGA1 (shA1)—three replicates per condition. Source data are provided as a Source Data file. **b** Gene expression log-fold changes (logFC) in OIS compared to Grow and OIS-shA1 compared to OIS of the genes differentially expressed in both comparisons, clustered by the direction of the change. **c** Immunofluorescence imaging of Grow, OIS and OIS-shA1 stained for DAPI (nuclear; blue), IL-8 (green), and HMGA1 (magenta) cells (n = 2 per condition, quantification—Supplementary Fig. 1f, Source data are provided as a Source Data file. **d** Overlap between HMGA1 consensus peaks in the IMR90 Grow and OIS and H1299 cells. **e** Correlation between AT% and average (log) HMGA1 signal in 200 kb bins. **f** Top: ChIP-seq normalised signal tracks of HMGA1, H3K9me3 and Lamin B1 in the Grow and OIS conditions (IGV browser); Bottom: close-up view of the HMGA1 peaks in a smaller genomic region. **g** Overlaps (genomic area covered in Mb) between HMGA1-dense regions, Lamin-associated domains (LADs), and H3K9me3 peaks in growing IMR90 cells. **h** Number of HMGA1 binding changes in OIS compared to Grow cells (Up—increased, Down—decreased). **i** Properties of the n = 16,420 peaks with increased (Up) and n = 4060 peaks with decreased (Down) HMGA1 binding from **h**: distance to the nearest HMGA1-dense region (left) and the nearest H3K9me3 peak (right). **j** ChIP-seq normalised signal tracks of HMGA1 and H3K27ac over representative genes with decreased HMGA1 on their gene body: *MMP1, CSF2, IL6*. **k** Distribution of log-fold changes in OIS compared to Grow and OIS-shA1 compared to OIS of the genes DE in each respective comparison classified as: 1) overlapping HMGA1-dense regions (n = 893 and 218 genes), 2) with any HMGA1 binding (n = 2580 and 518 including genes in HMGA1-dense regions), and 3) not bound by HMGA1 at all (n = 2756 and 445 genes), respectively. **j, k** Box plot centre line represents the median, the bounds correspond to the 0.25 and 0.75 quantiles, the whiskers represent the 0.1 and 0.9 quantiles. Significance testing was performed using two-sided Wilcoxon tests.

(e.g., *IFNAs* and *IFNB1*) (Supplementary Fig. 2g). The OR clusters are highly tissue-specific and, thus, heterochromatic in fibroblasts[17]. While *IFNB1* is a known direct HMGA1-target gene[24] and *IFNA* and *IFNB1* were previously shown to be induced in 'late senescent' fibroblasts[25], they were not expressed in OIS IMR90 cells at the time point we used (day 6).

HMGA1 was also previously implicated in the formation of the enhanceosome[26] and, in our data, 21% of all enhancers exhibited at least one HMGA1 peak in proliferating fibroblasts. We looked further at enhancers overlapping HMGA1-dense regions and found that they were similar between proliferating and OIS cells (83% match). HMGA1-dense enhancers were shorter in width and had lower levels of H3K27ac, a histone mark for active enhancers (Supplementary Fig. 3a), suggesting that HMGA1 may be generally linked to enhancers with reduced activity. The genes proximal to these enhancers showed enrichment for 'Epithelial Mesenchymal Transition', including the *LUM, FBN1, FAP* and *FGF2* genes, but also some cell cycle related genes such as *CCNA2* and *PLK4* (Supplementary Fig. 3b). Out of the 1,046 genes proximal to HMGA1-dense enhancers, 123 were differentially expressed in OIS-shA1 compared to OIS, suggesting a potential role for HMGA1 in their regulation.

Together, these data show that, although high-density HMGA1 peaks typically co-localize with classical heterochromatin and promoters are mostly depleted of HMGA1, a substantial number of peaks overlap enhancer or genic regions.

We next focused on differential binding analysis between Grow and OIS, which yielded 21,398 increases and 6676 decreases in HMGA1 binding (Fig. 1h). The respective genomic locations showed distinct features: peaks with increased HMGA1 were closer to HMGA1-dense regions and H3K9me3 peaks and more AT-rich than peaks with decreased HMGA1 binding (Fig. 1i, Supplementary Fig. 3c). We found 1094 differentially expressed (DE) genes with increased HMGA1 binding in OIS on the gene body and 460 DE genes with decreased binding. The DE genes with increased HMGA1-binding were enriched for cell cycle signatures (Supplementary Fig. 3d), including *CCNB1, CDK1* and *PCNA*, and tended to be down-regulated (62% of DE genes) during OIS. In contrast, while the DE genes with decreased HMGA1 binding included some SASP components, such as *MMP1, CSF2* and *IL6* (Fig. 1j, Supplementary Fig. 3d), reduced HMGA1 binding showed no overall direction of gene expression change.

To gain a better understanding of the effect of HMGA1 binding on gene expression, we cross-examined HMGA1 occupancy and HMGA1-responsive genes. The genes bound by HMGA1 and DE in OIS, particularly within HMGA-dense regions, were mostly down-regulated and this trend was largely alleviated with HMGA1 knock-down (OIS-shA1 compared to OIS), supporting that these regions become repressive during OIS in an HMGA1-dependent manner (Fig. 1k, Supplementary Fig. 3e). As mentioned earlier, HMGA1-dense regions showed good overlap with

H3K9me3, which may contribute to its repressive role. However, the HMGA1-dense H3K9me3 regions were mostly gene-poor and inactive, with only 44 out of a total of 764 genes expressed in IMR90 cells. In contrast, non-H3K9me3 HMGA1-dense regions overlapped with 2738 genes, 1595 of which were expressed in IMR90 cells. Thus, our data suggest that HMGA1-dense regions gain their gene repressive activity during OIS, even without direct deposition of the heterochromatic mark H3K9me3. In contrast, genes lacking HMGA1 binding tended to be up-regulated during OIS, and, interestingly, this was also HMGA1-dependent (Fig. 1k).

Thus, our data show that HMGA1 is linked to gene repression in HMGA1-dense regions and activation in regions lacking HMGA1 during senescence.

## HMGA1 promotes chromatin compartmentalization

Our data so far suggest that HMGA1 has a profound impact on both gene repression and activation during OIS, which is not easily explained by current models (e.g., heterochromatic repression; activation through direct binding to regulatory elements). HMGA1 plays a key role in the chromatin re-organization of senescent cells, contributing to SAHF formation, and therefore HMGA1 may affect gene expression by 3D repositioning. To interrogate HMGA1's role in 3D genome organization, we performed Hi-C experiments with shA1 in both Grow and OIS conditions (451 and 306 million reads, respectively, after removal of artefacts), matching our previously published Grow and OIS dataset[8]. The agreement between replicates was assessed using HiCRep[27] (Supplementary Fig. 3f).

Differential Hi-C interaction analysis at 200 kb resolution revealed that, in the OIS condition, HMGA1 depletion resulted in 18,500 changes, the majority of which involved HMGA1-bound regions (Fig. 2a, b, Supplementary Fig. 3g), indicating a direct role for HMGA1 in chromatin 3D re-organization. OIS is known to exhibit distal trans-compaction, which is thought to reflect SAHF formation[3,6-8]. Strikingly, the OIS-associated trans-compaction was largely reversed with shA1 (Fig. 2a, d), further reinforcing the essential role of HMGA1 in SAHF formation.

Consistent with a previous study linking OIS to stronger chromatin A/B compartmentalization[7], our OIS model also displayed stronger compartmentalization, marked by reduced A-B and increased A-A and distal B-B interactions (Fig. 2e, Supplementary Fig. 3h), despite the relative stability of A/B compartment classification during OIS in IMR90 cells[8]. While HMGA1 depletion had a limited impact on compartment switches (OIS-shA1 compared to OIS, corr. 0.98 between the first PCs), it largely reversed the enhanced compartmentalization during OIS, except with a further increase in A-A interactions (Fig. 2e). A weaker but similar trend was also observed in normal fibroblasts with sh-A1 (Grow-shA1) condition, with reduced B-B interactions and

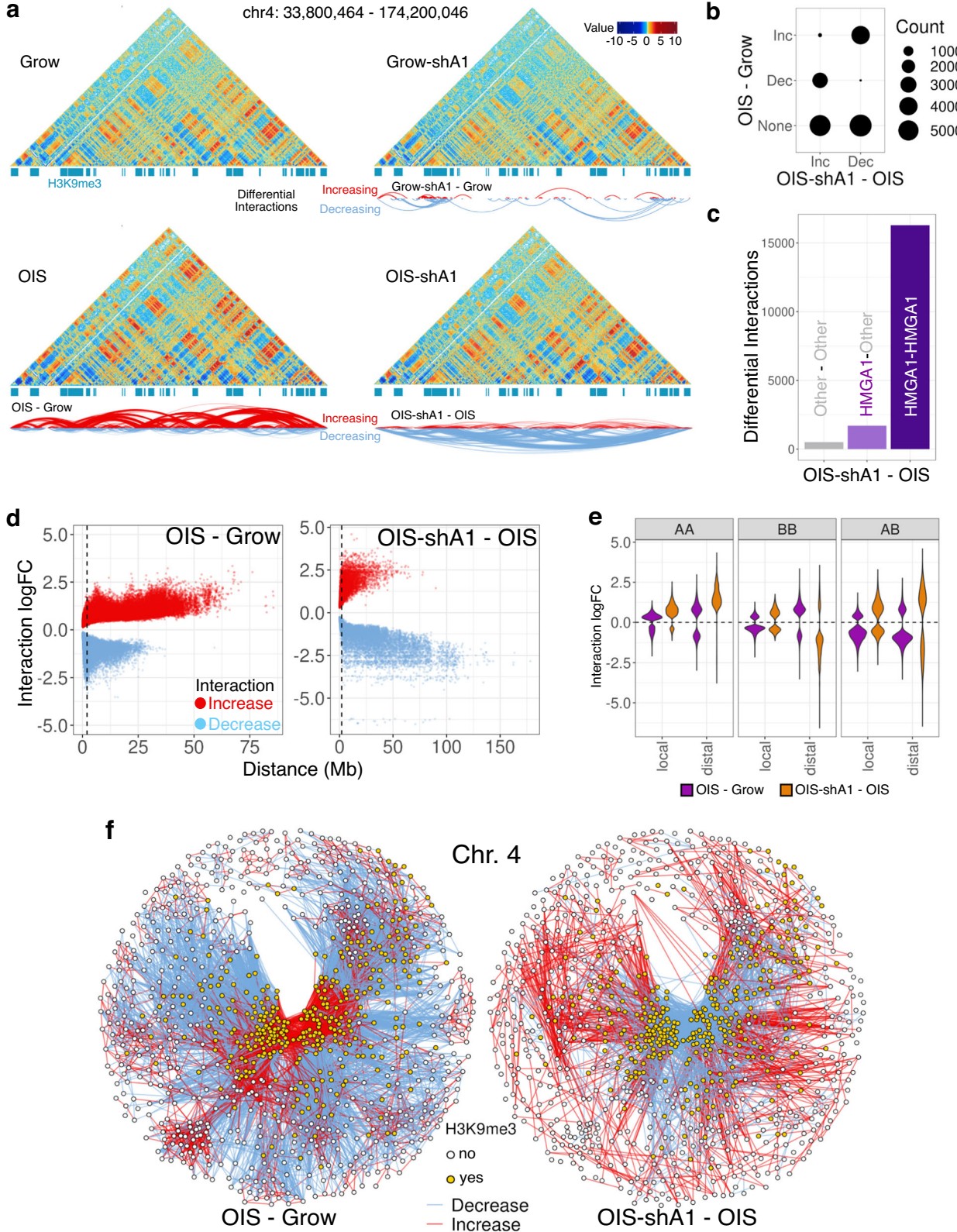

increased A-A interactions, suggesting a more general role of HMGA1 towards supporting chromatin compartmentalization and heterochromatin organisation (Supplementary Fig. 3i).

These results indicate that the degree of local HMGA1 deposition has a global impact on high-order chromatin status, possibly through coordinated effects on both HMGA1-rich and HMGA1-poor regions within chromatin networks.

## The *k*-core decomposition of the chromatin network

Previous studies linked SAHF formation exclusively to the increased distal contacts between heterochromatic regions, in a pairwise manner. Here, we characterized instead the global changes in the chromatin network using a graph-theoretical approach to identify densely connected clusters. We reasoned that this will not only reveal the nature of chromatin compaction, but also the global impact of OIS on

**Fig. 2 | HMGA1 leads to global re-organisation of chromatin architecture in OIS.**
**a** Hi-C interaction maps (resolution 200 kb) of Grow, Grow-shA1, OIS, and OIS-shA1
conditions, with H3K9me3 peaks marked (blue) and arcs representing increased
(red) and decreased (blue) interactions. **b** Number of significantly increased (Inc)
and decreased (Dec) interactions in OIS-shA1 compared to OIS, crosschecked
against the changes in OIS compared to Grow. **c** The number of OIS-shA1 compared
to OIS interaction changes where one or both regions involved is bound by HMGA1
(minimum 5 peaks per bin). **d** The distance between the regions involved in dif-
ferential interactions against the interaction log-fold changes in OIS compared to
Grow (left) and OIS-shA1 compared to OIS (right); the vertical dashed line marks the

2 Mb distance threshold denoting distal interactions. **e** Log-fold changes of the
differential interactions between A-compartment regions (AA), B-compartment
regions (BB) and between A- and B-compartment regions (AB), categorised by local
(within 2 Mb) and distal (>2 Mb) interactions, in the OIS compared to Grow (purple)
and OIS-shA1 compared to OIS (orange) comparisons. **f** Differential interaction
networks of chromosome 4 between nodes representing 200 kb bins, coloured by
H3K9me3 status, and edges representing increased (red) and decreased (blue)
interactions; left: OIS compared to grow and right: OIS-shA1 compared to OIS;
nodes are positioned according to the Fruchterman-Reingold layout calculated
based on the edge weights representing the increased interaction log-fold changes.

the interactions network, including effects on non-heterochromatic
regions.

We constructed networks for each chromosome using genomic
bins as nodes and differential interactions between conditions as
edges (Fig. 2f). In OIS, we found large clusters of increased interac-
tions, which were disrupted in the presence of shA1, consistent with
SAHF formation (Fig. 2f). The highly interacting clusters often corre-
sponded to HMGA1-dense regions overlapping H3K9me3, but
H3K9me3-independent clusters were also present (e.g., chromosome
1, Fig. 3a). HMGA1-dense regions without H3K9me3 also organized
around the densely interacting clusters. Interestingly, we also
observed regions which were excluded from these clusters, marked by
many decreased interactions, during OIS, in an HMGA1-dependent
manner.

Several measures exist for analysing the connectivity of a graph:
e.g., the degree distribution or centrality measures. To capture
dynamic nature of chromatin networks, we applied the $k$-core
decomposition[28] to the differential Hi-C connectivity maps (Fig. 3b, c):
it identifies dense subgraphs, i.e., highly interconnected subsets of the
network (relative to the number of nodes), allowing for both intuitive
and quantitative assessment of network structure (see Supplementary
Information).

We first focused on the networks of increased interactions during
OIS to determine the $k$-core values of the genomic bins (nodes). The $k$-
core values correlated overall with epigenetic properties, capturing
distinct chromatin units (Fig. 3f): Regions with a high $k$-core (>5) were
enriched for HMGA1, H3K9me3, and depleted of Lamin B1 in OIS, likely
reflecting SAHF formation. Regions with medium $k$-core (3–5) tended
to retain Lamin B1 in OIS cells. These regions overlapped with
H3K9me3, but not with HMGA1, possibly corresponding to residual
perinuclear heterochromatin in OIS cells. Low $k$-core (<3) was mostly
associated with euchromatic features, marked by enrichment in
H3K27ac and ATAC-seq signal.

We analysed the degeneracy cores of each chromosome (bins
with maximal $k$-core in the networks of increased interactions) and
found that chromosomes 1–14 and 18 had prominent cores, with $k$-max
over 10 (Supplementary Fig. 4a), while chromosomes with high gene
density, such as 17, 19 and 22, had low $k$-max values. Interestingly, $k$-
max values strongly correlated with HMGA1 binding, more than with
other features, including H3K9me3 (Supplementary Fig. 4a, b). The
chromosomes with low $k$-max values indeed had lower AT% distribu-
tions and fewer HMGA1 peaks (Supplementary Fig. 4d).

Chromosome 1 had the most prominent degeneracy-core, with $k$-
max equal to 32 (Fig. 3c). This subgraph corresponded to the cluster
identified earlier as being highly enriched for HMGA1 but depleted of
H3K9me3 (Fig. 3a, d), highlighting the central role of HMGA1 in
forming these densely interacting clusters. The striking aggregation of
these HMGA1-dense regions on chromosome 1 during OIS suggests
that HMGA1 alone can contribute to chromatin compaction. Some
chromosomes (e.g., chromosomes 1 and 9) had a prominent dense
subgraph on each arm, while the degeneracy-core of most other
chromosomes spanned both chromosome arms, reflected by distal
interaction increases between the two arms during OIS. We also found

other dense subgraphs with relatively high $k$-core values: chromosome
3 had a secondary dense subgraph which was marked by H3K9me3 but
lower HMGA1 than its degeneracy core (Fig. 3e), reinforcing the link
between high $k$-core values and HMGA1. Moreover, this secondary
cluster retained Lamin B1, whereas the main, HMGA1-dense cluster
exhibited Lamin B1 disruption (Supplementary Fig. 4c), reflecting our
previous observation regarding the correlation between Lamin B1 loss
and HMGA1 binding (Supplementary Fig. 2c).

We incorporated decreased connectivity in our $k$-core analysis to
classify genomic regions involved in differential interactions (Fig. 3g).
Regions with high $k$-core values (min. $k$-core 6) were defined as Core
and regions with increased interactions with the Core were classified as
Peri. Smaller clusters (min. 4) which were independent or detached
from Core regions were classified as alternative cores, AltCore. All
regions excluded from Core and AltCore (decreased interactions) were
classified as ExCore.

In terms of A/B compartments, Core and Peri regions mostly
corresponded to the B-compartment (Fig. 3h, negative AB score), while
AltCore and ExCore regions were associated with both A and B com-
partments, and Other regions were mostly in the A compartment
(positive AB score). We also reanalysed the (epi)genetic features
according to these classes (Fig. 3h–j): Core regions had the highest
levels of HMGA1 binding, were highly heterochromatic (marked by
H3K9me3, highest AT% and lowest gene density) and lost Lamin B1 in
OIS. Peri regions had lower HMGA1 and H3K9me3 than Core. AltCore
regions were enriched for H3K9me3 but had low levels of HMGA1 and
tended to retain Lamin B1 in OIS. They were also more euchromatic
than Core and Peri regions, marked by higher ATAC-seq and H3K27ac
levels. ExCore regions were mostly euchromatic (low AT%, gene dense
and high ATAC and H3K27ac signal), with low H3K9me3, but not as
euchromatic as the unclassified, 'Other' regions (Fig. 3h–j).

HMGA1 substantially affected the inter-connectivity of these
classes: the increased interactions forming the Core and Peri-Core
connections in OIS were cancelled in OIS-shA1 (Fig. 4a). These regions
may collectively contribute to SAHFs. Notably, the exclusion of Alt-
Core and ExCore regions from Core was also largely HMGA1-
dependent (Fig. 4a), supporting the coupling of direct and indirect
roles of HMGA1 to shape global chromatin environment.

## Architectural role of HMGA1 in gene regulation during senescence

Finally, we characterized the genes and their expression changes (in
either OIS compared to Grow or OIS-shA1 compared to OIS) in each of
the classes. Overall, the genes within Core, Peri, and AltCore regions in
OIS cells had a down-regulation tendency, which was reversed by
HMGA1 depletion, while genes excluded from cores (ExCore) were
more likely to be up-regulated during OIS and further up-regulated
with shA1 (Fig. 4b). This suggests that incorporating genes in hetero-
chromatic cores may contribute to their down-regulation in an
HMGA1-dependent manner and excluding them from cores may con-
tribute to their up-regulation in OIS.

As expected, due to their highly heterochromatic features, genes
within Core regions consisted of many gene families known to be

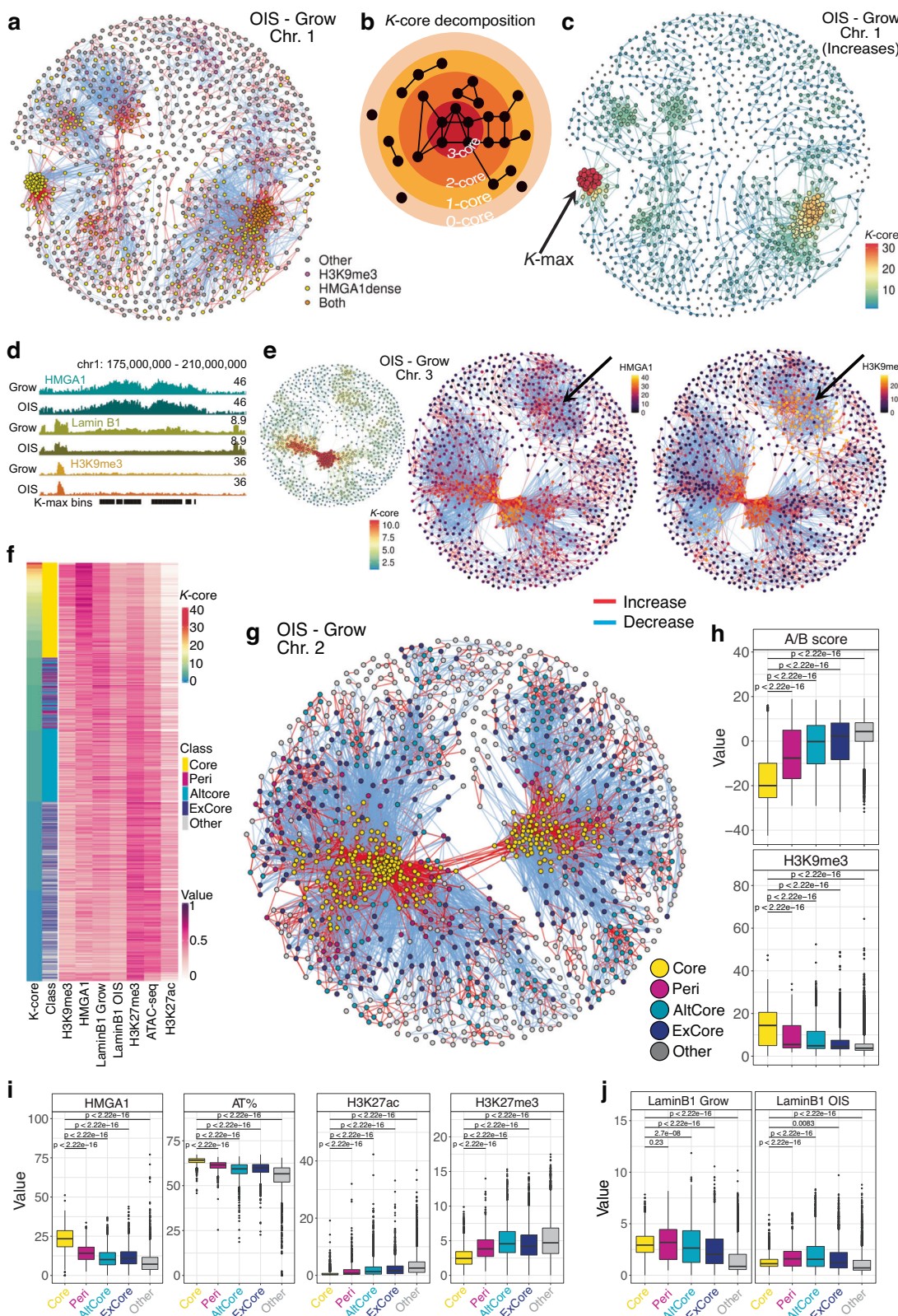

repressed in most tissues, such as olfactory receptors and the type-I interferon genes (Supplementary Fig. 4e). While promoters of these genes were often HMGA1-bound (Supplementary Fig. 2g), the involvement in Cores might explain why we did not observe *IFNB1* induction, which has been implicated in senescence[25], in our experimental model, despite the abundant evidence for HMGA–mediated activation of *IFNB1*[11]. Interestingly, we found that Core regions without H3K9me3

(HMGA1-dense only) were mostly responsible for the trend of HMGA1-dependent down-regulation during OIS (Fig. 4c), as the Core regions with H3K9me3 overlapped very few expressed genes.

Core down-regulated genes reflected cell-cycle signatures, including genes such as *CCNA2* and *CDK1* (Fig. 4d, f). This is reminiscent of the incorporation of cell cycle genes into SAHFs, as has long been postulated[9,29]. They were also enriched for epithelial-

**Fig. 3 | Classification of genomic regions based on their differential interactions connectivity patterns. a** Differential interactions network of OIS compared to Grow on chromosome 1 with nodes coloured by their overlap with H3K9me 3 and HMGA1-dense regions. **b** Model representation of the $k$-core decomposition of an example network with $k$-max equal to 3. **c** The $k$-core decomposition of the network of increased interactions in OIS compared to Grow on chromosome 1, with the size of each node (200 kb bins) reflecting the degree (number of interactions) of the node and the $k$-core value as the node colour; edge colour reflects the minimum $k$-core of its nodes; arrow indicates the nodes with the maximum $k$-core value ($k$-max). **d** Normalised ChIP-seq signal tracks of HMGA1, Lamin B1, and H3K9me3 in Grow and OIS around the regions which form the degeneracy core ($k$-max) of chromosome 1 shown in **c**. **e** The $k$-core decomposition (left) of the chromosome 3 network (200 kb bins) of increased contacts in OIS compared to Grow, with the node colour corresponding to its $k$-core and the node size to its degree; the full network of differential interactions highlighting the HMGA1 (middle) and H3K9me3 (right) normalised ChIP-seq signal of each node; arrow indicates the cluster of regions with low $k$-core and HMGA1, but high H3K9me3 ChIP-seq signal. **f** Heatmap of the $k$-core values, classification, and epigenetic properties, represented by the ChIP-seq (or ATAC-seq) scaled normalised signal of H3K9me3, HMGA1, LaminB1 in Grow and OIS, H3K27me3, ATAC-seq (accessibility), and H3K27ac, of all the bins with $k$-core of at least 1. **g** The network of (all) differential interactions of OIS compared to Grow on chromosome 2 with nodes classified as: Core (yellow), Peri (peri-core, magenta), AltCore (alternative core, cyan), ExCore (excluded from cores, navy blue) and other (grey). **h** The A/B compartment score and H3K9me3 ChIP-seq signal of the genomic bins in each of the classes. **i** The distributions of the HMGA1 signal, the AT%, H3K27ac and H3K27me3 signal of the bins in each class. **j** Lamin B1 signal in Grow (left) and OIS (right) of all the bins in each of the classes. **h–j** Compared regions: n = 2900 Core, n = 915 Peri, n = 2892 AltCore, n = 2368 ExCore, and 5235 Other. Box plot centre line represents the median, the bounds correspond to the 0.25 and 0.75 quantiles, the whiskers represent the 0.1 and 0.9 quantiles. P-values derived from two-sided Wilcoxon testing.

mesenchymal transition (EMT), which included many commonly used fibroblast markers, such as *COL1A2, DCN, LUM, VCAN* and *FAP*[30]. It has been shown that senescent fibroblasts tend to lose fibrogenic activities and our data provide a mechanistic insight into the senescence-associated shift of cell identity[1]. Notably, these genes were not necessarily marked by H3K9me3, but rather they were within a high HMGA1 binding environment, which led to their incorporation into the dense H3K9me3 Cores (Fig. 4d). Other down-regulated genes, including another cell cycle gene, *PCNA*, were included in Peri regions with increased interactions with Core regions. Genes down-regulated within AltCore regions were also enriched for cell cycle regulators (e.g., *RB1* and *BUB1*), but also for TGFβ signalling (e.g., *TGFB1* and *TGFBR1*) (Fig. 4d). On the other hand, up-regulated genes excluded (ExCore) from the Cores and AltCores included *CDKN2A* (p16) and *CDKN2B* (p15), as well as key SASP factors, such as *IL1, MMP1,3* and *CXCL1,2,3* (Fig. 4e, f).

So far, we assessed the impact of HMGA1 (via shA1) on the chromatin re-organization of OIS (relative to Grow), which consists of mostly 'cancelling' the connectivity changes occurring in OIS−Grow, with very few changes in the same direction, e.g., increased in OIS and in OIS-shA1. However, we also found substantial interaction changes in OIS-shA1 from OIS which did not change during OIS (Fig. 2b). These de novo changes reflected the higher A-A and A-B distal contacts in OIS-shA1 relative to OIS (Fig. 2e). Indeed, de novo decreased interactions in OIS-shA1 consisted mostly of B-B pairs (Fig. 4g), suggesting the enhanced disconnection of heterochromatic regions with shA1. In contrast, the de novo gained interactions in OIS-shA1 mostly consisted of A-A interactions, demonstrating an increased connectivity between euchromatic regions (Fig. 4g). 71% of these increases were potentially regulatory, corresponding to contacts between gene promoters and enhancers, involving a substantial proportion of the DE genes in OIS-shA1 relative to OIS (632 genes). These genes were enriched for NF-kB signalling and inflammatory response, including the *MMP* and *IL1* genes (Fig. 4h), providing a potential mechanism for the further up-regulation of the inflammatory SASP in OIS-shA1. One notable example was the contact between the *CXCL* genes (including *CXCL2* and *CXCL8/IL8*) and a distal enhancer situated 5 Mb away (Fig. 4i), which was bound by HMGA1 in OIS, potentially hindering the interaction with the *CXCL* locus in the presence of HMGA1. HMGA1-dense enhancers, which appeared to be less active (Supplementary Fig. 3a), indeed showed lower values of average connectivity than other enhancers in OIS cells (Fig. 4j), and HMGA1 depletion promoted their connectivity in the OIS context, potentially contributing to its buffer activities in gene regulation (Fig. 4k).

Together, this evidence suggests that HMGA1 plays an important role in chromatin organization by promoting compartmentalization and modulating the potency of enhancers. The impact of HMGA1 on global chromatin configuration, rather than local binding of

individual gene promoters, may represent an additional layer of gene regulation.

## HMGA1-driven transcriptional programmes at single-cell level

Our data suggest a profound impact of HMGA1 on chromatin architecture, and consequently, on the senescent transcriptome. However, these conclusions are based on bulk experiments that represent the average changes over millions of cells and do not take into account the reported heterogeneity of the senescent phenotype[2]. To better understand how the gene expression is modulated by HMGA1, we analysed the senescence transcriptome (with and without shA1) at single-cell level in IMR90 cells (Fig. 5a). After filtering low quality cells, the final dataset consisted of around 11,000 cells. Principal component analysis showed good agreement between replicates and as expected, HMGA1 expression levels increased in OIS and decreased with shA1 (Supplementary Fig. 5a, b).

We confirmed the robustness of the senescent transcriptome by checking the down-regulation of cell-cycle markers (e.g., *E2F1, CCNA2, MKI67*) and the up-regulation of key SASP factors (e.g., *IL1B* and *IL8*). Notably, these senescence markers followed interesting patterns across the cells, with substantial heterogeneity (Fig. 5a). We also computed an enrichment score for the Hallmarks Signatures in The Molecular Signatures Database[31] (MSigDB) (Fig. 5b) and observed that E2F targets were largely repressed in OIS compared to Growing cells, but less so in OIS-shA1 cells. This is consistent with our previous data, showing OIS-shA1 cells are more susceptible to senescence escape than OIS cells with simultaneous knock-down of p16[13]. The enrichment of NF-kB signalling in OIS was further enhanced with shA1, consistent with the bulk-RNA-seq results (Fig. 1b, Supplementary Fig. 1c, d). Other notable signatures included p53 Pathway and Notch signalling genes, which were more enriched in OIS and less so in OIS-shA1 cells (Fig. 5b).

We used the Milo[32] statistical testing method to characterise the shift in OIS-shA1 cells and partial overlap with the HRAS^G12V-expressing cells, consistent with their maintained senescence-like status[13]. We determined which senescence sub-populations are under- or over-represented upon HMGA1 depletion and found 4 clusters: two over- (Clusters 1 and 3) and two under-represented (Clusters 2 and 4) in OIS (Fig. 5c).

Both Clusters 1 and 2 reflected typical senescence features, but to different degrees. While both lacked expression of cell cycle genes, Cluster 1 (low-HMGA1) exhibited more prominently the inflammatory SASP (e.g., *IL6, IL1B, MMP1, MMP3*), and was strongly depleted of fibrogenic features (Fig. 5d, e, Supplementary Fig. 5c). Thus, Cluster 1 most likely represents the senescence-associated cell identity and functionality shift[1], which has been described based on whole-population assays (Fig. 1a, b). While Cluster 2 expressed the highest levels of *HMGA1* and *CDKN2A/p16*, the inflammatory SASP was modest

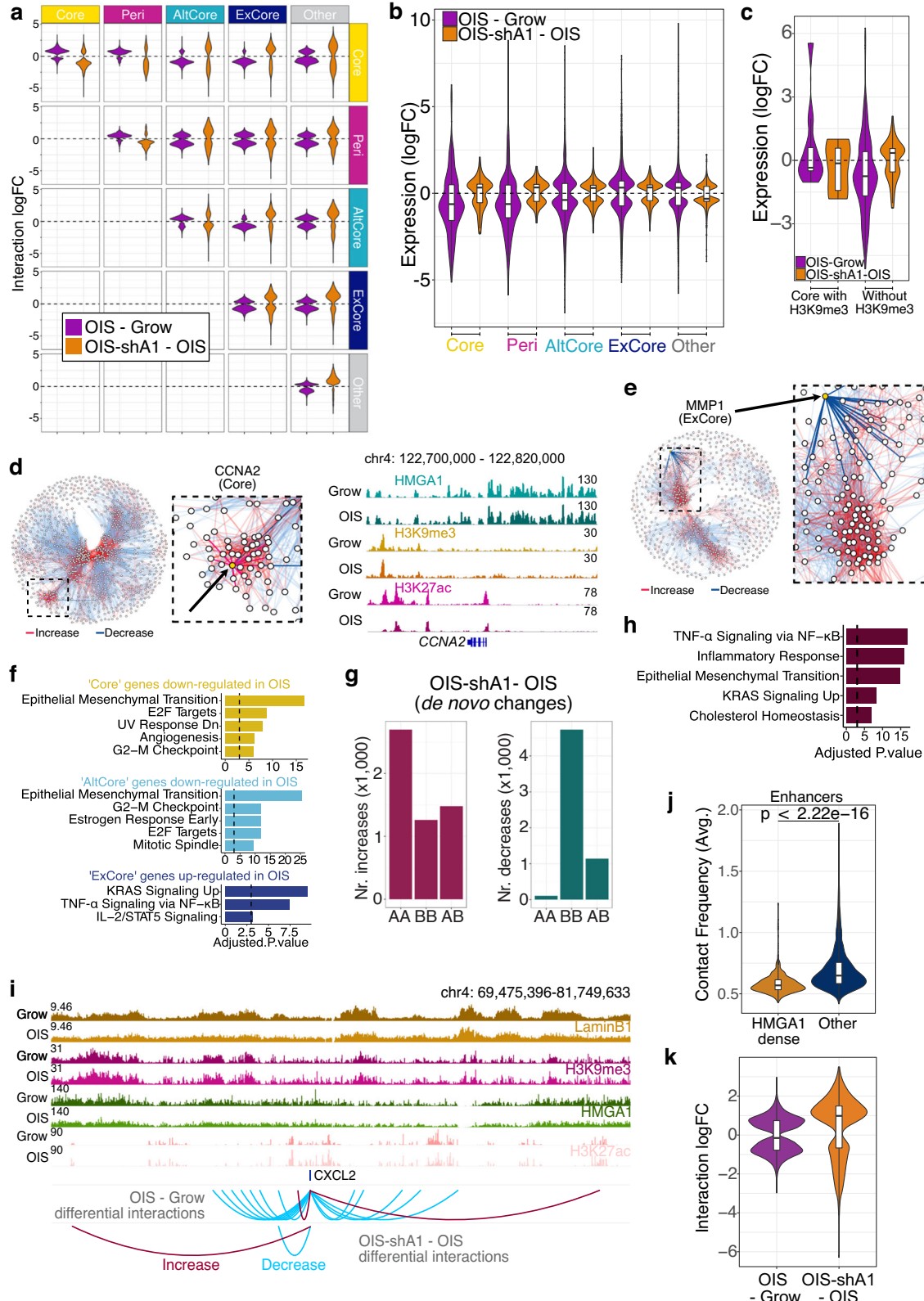

but instead expressed other secretory/cell surface components (Fig. 5d, e, Supplementary Fig. 5d). Therefore, HMGA1 may contribute to fine-tuning between these senescence states.

Clusters 3 and 4 retained fibroblastic features (represented by EMT signature), but more prominently in Cluster 4 (e.g., *FN1*, *SPARC* and *MYLK*), with little expression of the inflammatory SASP components (Fig. 5d, Supplementary Fig. 5e, f). This phenotype is highly

reminiscent of NOTCH-mediated secondary senescence[33,34]. We previously showed that NOTCH-induced senescence (NIS) represents an early phase of senescence, characterised by enhanced fibrogenic and reduced inflammatory features[33]. Our current results suggest that HMGA1 plays a role in the regulation of NIS-like dynamics during senescence, reinforcing that NIS represents a distinct feature with a unique molecular signature as an integral part of senescence.

**Fig. 4 | The effects of HMGA1 loss on the chromatin interactions network and gene expression. a** The log-fold changes of the differential interactions between the regions in each pair of classes in OIS compared to Grow (purple) and OIS-shA1 compared to OIS (orange). **b** The distributions of the gene expression log-fold changes in the OIS compared to Grow and OIS-shA1 compared to OIS comparisons of the genes within the five classes which are also DE in their respective comparisons (n = 253 and 130 genes in Core, n = 225 and 98 genes in Peri, n = 1015 and 398 genes in AltCore, and n = 1088 and 418 genes in ExCore, and n = 4333 and 1411 genes in Other). **c** Gene expression log-fold changes in OIS compared to Grow and OIS-shA1 compared to OIS of the respective DE genes within 'Core' regions with (n = 9 and 6 genes) and without H3K9me3 (n = 208 and 101 genes). **d** The position of the Core *CCNA2* gene (left) in the networks of differential interactions on chromosomes 4 in OIS compared to Grow and the ChIP-seq signal of HMGA1, H3K9me3 and H3K27ac at the *CCNA2* locus (right). **e** The position of the ExCore gene *MMP1* in the chromosome 11 network of differential interactions in OIS−Grow. **f** The top MSigDB Hallmarks gene sets enriched in the genes down-regulated (in OIS compared to Grow) and overlapping 'Core' and 'AltCore' regions, and the genes up-regulated (in OIS compared to Grow) and overlapping the 'ExCore' regions. **g** The increased (left)

and decreased (right) de novo interactions in OIS-shA1 compared to OIS, according to the A- or B- compartment assignments of the interacting regions. **h** Gene set enrichment analysis against the MSigDB Hallmarks of the genes DE in OIS-shA1 compared to OIS and involved in the de novo increased interactions in the same comparison. P-values derived with EnrichR−Fischer's exact test adjusted for multiple comparisons with Benjamini-Hochberg. **i** ChIP-seq normalised signal tracks of Grow and OIS Lamin B1, H3K9me3, HMGA1, and H3K27ac of the *CXCL2* gene locus and the regions involved in increased (red) and decreased (blue) interactions in the OIS compared to Grow and OIS-shA1 compared to OIS, respectively. **j** The average contact frequencies in OIS of the enhancers within HMGA1-dense regions (n = 833 bins overlapping these enhancers) compared to other enhancers (n = 2610 bins overlapping these enhancers). **k** The interaction log-fold changes in OIS compared to Grow and OIS-shA1 compared to OIS of only the enhancers within HMGA1-dense regions (n = 833 bins overlapping these enhancers). **b, c, j, k** Box plot centre line represents the median, the bounds correspond to the 0.25 and 0.75 quantiles, the whiskers represent the 0.1 and 0.9 quantiles. P-values derived from two-sided Wilcoxon testing.

In contrast, Cluster 3 (low-HMGA1) substantially expressed cell cycle genes (Fig. 5d), suggesting that these cells are the major source of the unstable senescence feature observed in OIS-shA1 cells at population-level[13].

Our data show that HMGA1 contributes to the dynamic heterogeneity of senescence, by regulating genes involved in key signatures, i.e., cell proliferation, inflammatory SASP, and cell identity (fibroblast/EMT) markers (model−Fig. 5f).

### The interplay between HMGA1 and key senescence TFs mediates the senescence spectrum

The enrichment of the genes affected by HMGA1 for NF-kB signalling (de novo interactions−Fig. 4h and ExCore−Fig. 4f) suggests that HMGA1 may refine transcription factor activity by modulating the chromatin environment. Consistently, HMGA1-responsive genes showed enrichment for gene sets representing direct targets of TFs active in OIS (e.g., NF-kB, C/EBPβ, p53, Supplementary Fig. 5g).

To experimentally validate whether HMGA1 acted in conjunction with other known TFs during OIS, we compared its transcriptional effect to those of C/EBPβ (a key TF responsible for inflammatory SASP[35]) and p53[36]. For this, we stably expressed shCEBPB or shp53 in ER:HRAS^G12V IMR90 cells, then oncogenic RAS was induced by 4OHT as in the OIS-shA1 setting. We also included the transcriptome of OIS with a double knock-down of C/EBPβ and p53 due to their complex interplay and potential synergistic effect[36].

As expected[33,35,37], the C/EBPβ depletion blunted the inflammatory SASP after oncogenic RAS induction, but it also restored some fibroblast (EMT) markers, mirroring HMGA1 depletion in the OIS condition (Supplementary Fig. 6a, b). Interestingly, the effects of both C/EBPβ and p53 depletion were mostly potentiated by the double knock-down (Supplementary Fig. 6c−e), reinforcing their cooperative activity during OIS.

We applied dimensionality reduction (principal component analysis) to the log-fold changes in response to all the knock-down experiments of HRAS^G12V-expressing cells and found that HMGA1 acted in the opposite direction of the C/EBPβ and p53 activity (Fig. 5g). Furthermore, at the single-cell level, we used the signature of the genes activated by the combined effect of these TFs to calculate an enrichment score (C/EBPβ + p53) for the OIS cells and found a striking overlap with the HMGA1-repressed gene signature during OIS (HMGA1-buffered genes), suggesting that HMGA1 depletion in the OIS context enhances the activity of these TFs (Fig. 5h). Indeed, based on our previous ChIP-seq datasets, the C/EBPβ + p53 signature was enriched for direct targets of these TFs[17,36]. These data support that HMGA1 can modulate TF activities through global chromatin architectural alteration.

### Buffering effect of HMGA1 on the pro-inflammatory signature in cancer

HMGA1 up-regulation is frequently observed in many cancer types, including lung cancer, and linked to poor survival[38]. We tested the transcriptional effect and the chromatin binding of HMGA1 in the H1299 NSCLC cell line. The transcriptional changes in H1299 cells stably expressing shA1 revealed a similar alteration as in IMR90 fibroblasts, consisting of the up-regulation of pro-inflammatory genes (Fig. 6a) and the down-regulation of Notch signalling genes, such as *NOTCH1*, *JAG2* and *MGP*. Moreover, the HMGA1 binding profile was conserved between the H1299 and IMR90 cells (Fig. 1d), with 232,290 common peaks and while peak calling identified more IMR90-specific peaks, only ~10,000 regions were completely devoid of HMGA1 binding in H1299, compared to IMR90 cells.

The similar gene expression changes and protein binding profile of HMGA1 observed in H1299 cells open exciting avenues for investigating the impact of HMGA1 in cancer. As HMGA1 binding correlated with AT-content, the binding of specific enhancers and genes may reflect their sequence features and may lead to the same genes being affected across different biological contexts. We took advantage of the recent advances in single-cell transcriptomics and compared the profiles of cells with high and low *HMGA1* levels (as a proxy for protein level) in publicly available single-cell datasets. First, we analysed the single-cell profiles of H1299 cells[39] and found that cells with low *HMGA1* were more likely to express genes involved in pro-inflammatory signalling, such as *CXCL1*, *CXCL2*, and *IL6*, compared to cells with high *HMGA1* (Fig. 6b, c). We also checked the transcriptomic associations of *HMGA1* in a human lung cancer dataset[40], which analysed multiple stages of lung cancer. We found that *HMGA1* is expressed in a higher proportion of epithelial cells in the advanced stages, compared to normal lung samples (Fig. 6d). Moreover, cells with high and low *HMGA1* also showed a polarised expression of pro-inflammatory genes, which were more likely to be expressed in the low *HMGA1* cells (Fig. 6e, f), suggesting that HMGA1 may buffer the pro-inflammatory phenotype in human cancers.

### Discussion

Important macrodomains, such as the A/B compartments, have been identified from Hi-C experiments. Our *k*-core analysis focuses on differential chromatin networks, defining distinct sub-networks, which reflect A/B compartments (Fig. 3h). Although HMGA1 had a limited effect on A/B compartment shifts, it showed the capacity to strengthen compartmentalization. Some of the high *k*-core regions lacked H3K9me3, uncovering a unique chromatin environment, driven by HMGA1. Genes in these Core-type regions with no H3K9me3 were typically repressed during OIS.

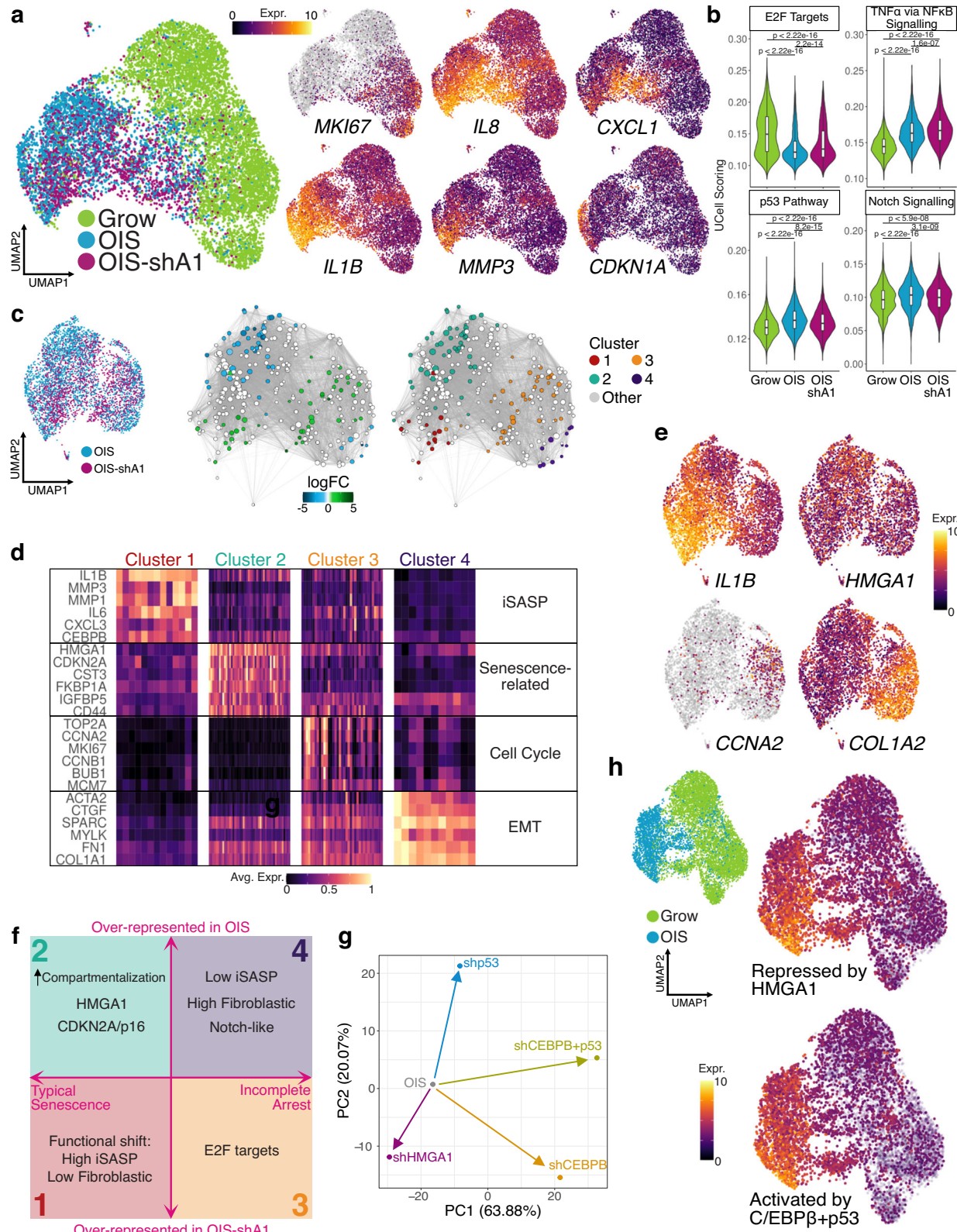

Thus, high density of HMGA1 on the linear genome is not necessarily repressive (e.g., in proliferating IMR90 cells), but the increased interaction density creates a repressive environment. Since these Core regions with no H3K9me3 tended to merge into H3K9me3-Cores, it remains to be elucidated whether HMGA1-mediated chromatin condensation is sufficient for gene repression, or the heterochromatin proximity is necessary.

Our chromatin network analysis revealed several HMGA1-driven gene regulatory mechanisms: i) gene repression in cores, ii) gene activation in regions excluded from cores, and iii) buffer activity of active genes outside of the cores. Although we do not exclude typical gene regulation via direct binding[41], the global impact of HMGA1 by modulating the chromatin environment appears to be more prominent. Despite the good correlation

**Fig. 5 | Senescence transcriptional programmes at single-cell level. a** Single-cell UMAP projection of Grow, OIS and OIS-shA1 cells (coloured by condition but each representing two replicates), highlighting key senescence genes: *MKI67*, *IL8*, *CXCL1*, *IL1B*, *MMP3*, and *CDKN1A*; expression values (log-transformed) were scaled to be between 0 and 10 for visualisation, with grey representing no expression detected. **b** Distribution of UCell single-cell scores for selected MSigDB Hallmarks gene sets in n = 6165 Grow, n = 2828 OIS, and n = 2073 OIS-shA1 cells; P-values derived from two-sided Wilcoxon testing. Box plot centre line represents the median, the bounds correspond to the 0.25 and 0.75 quantiles, the whiskers represent the 0.1 and 0.9 quantiles. **c** UMAP projection of the OIS and OIS-shA1 conditions only for Milo testing of cell neighbourhoods and clustering based on the log-fold changes between OIS and OIS-shA1. **d** Representative markers of the four Milo clusters of differential expression between OIS and OIS-shA1 cells at single-cell level, with expression values scaled between 0 and 1 and averaged over the cell neighbourhoods in each cluster from **c. e** Representative markers for clusters 1–4 coloured by

scaled expression values on the UMAP projection of OIS and OIS-shA1 cells. **f** Schematic representation of the features of the four clusters of senescent cells identified using overrepresentation analysis, highlighting the clusters over-represented in OIS (2 and 4) and in OIS-shA1 (1 and 3), respectively. Although inflammatory SASP (iSASP) and p16 have been collectively considered senescence hallmarks, they represent distinct types of senescence at the single-cell level. Clusters 3 and 4 express cell-cycle genes and Cluster 4 resembles the previously described NOTCH-related 'early phase senescence' with augmented fibroblastic features[33,34]. **g** Dimensionality reduction (PCA) of the log-fold changes of OIS cells (bulk RNA-seq) in response to shHMGA1, shCEBPB, shp53 and double knock-down of *p53* and *CEBPB*. **h** UMAP projection of the Grow and OIS cells, coloured by UCell scoring of the gene signatures of the genes activated in OIS and up-regulated by shA1 (repressed by HMGA1, top) and down-regulated by the double knock-down of *p53* and *CEBPB* (activated by *p53* + *CEBPB*, bottom).

between 3D environments driven by HMGA1 binding density and gene expression changes, we cannot exclude potential indirect effects of HMGA1 on transcription. We propose that the balance between the three identified mechanisms contributes to senescence heterogeneity (Fig. 5f). Cluster 2 in our single cell analysis corresponds to the highest HMGA1 level, possibly associated with enhanced compartmentalization, with the incorporation of cell cycle genes into Cores (this is reversed in Cluster 3). We speculate that these cells are SAHF positive. Indeed, this cluster expressed highly p16, a major senescence marker that promotes SAHF formation and senescence stability[13,42]. Reduced HMGA1 level is associated with two distinct states, OIS with enhanced SASP (Cluster 1) and incomplete arrest (Cluster 3). The former reflects the buffer effect of HMGA1 on the SASP. Thus, while HMGA1 contributes to senescence establishment, if cells are already committed to senescence, a reduced level of HMGA1 reinforces the functional shift. These data indicate an uncoupling of the key senescence markers, p16 and the SASP, further reinforcing the critical role of HMGA1 in senescence heterogeneity. Cluster 4 is unique, resembling NOTCH-induced senescence (NIS), which is characterized by enhanced fibrogenic features and reduced SASP[33,34]. NIS-like phenotype has been found at an early phase during senescence, consistent with the weak expression of cell cycle genes in this cluster. Notably, we previously showed that enforced activation of NOTCH signalling represses HMGA1 and SAHF formation[16]. Consistently, NOTCH-associated secondary senescence has been linked with lower *HMGA1* levels compared to primary senescence[43]. These data suggest a complex feedback loop between HMGA1 and NOTCH signalling.

The role played by HMGA1 towards the diversity of the senescent phenotype, including its buffer effect on the inflammatory signature, may also be relevant in the cancer context, where cellular and HMGA1 heterogeneity may contribute to shaping the inflammatory niche, since the buffer effect on pro-inflammatory genes appears to be conserved.

HMGA1-driven chromatin attraction/repulsion was more prominent in OIS, but a similar tendency was detected in the proliferating condition. Thus, our data may represent a general feature of chromatin configuration. The reason behind the strengthening of this feature during senescence is unclear. It is possible that the overall increase in the chromatin bound by HMGA1 promotes this process. We also speculate that additional factors, such as specific binding partners or post-translational modifications of HMGA1 might play important roles. This is consistent with the phase separation potential of HMGA1[44] and its close relative HMGA2[45]. Elucidating the multivalent aspect of its phase separation properties in diverse settings may provide mechanistic insights into fundamental and context-dependent impacts of HMGA1.

## Methods

### Cell culture and retroviral infection
IMR90 human diploid fibroblasts (CCL-186) and H1299 human non-small cell lung cancer (NSCLC) cells (CRL-5803) were obtained from the American Type Culture Collection (ATCC). IMR90 and H1299 cells were cultured in DMEM (Gibco #12481-023) with 10% fetal calf serum (FCS) at 37 °C in 5% (IMR90) or atmospheric (H1299) $O_2$ and 5% $CO_2$. Retroviral gene transfer was performed as described[46]. Briefly, retroviruses were produced in Phoenix-AMPHO packaging cells (ATCC) using calcium phosphate-mediated transient transfection of retroviral vector plasmids. Two days later, the supernatant containing the viral particles was filtered through a 0.45 μm filter. Cells were incubated with viral solutions in the presence of 4 μg/mL of polybrene. The filtered viral solutions containing 4 μg/mL of polybrene were added to cells. On the following day, the cells were placed in a fresh medium. After 24 hours of recovery with the fresh medium, cells were split into 1:2 with antibiotic selection (G418 (ThermoFisher, 10131035) 300 μg/mL for 7 days or puromycin (Merck, P9620) 1.5 μg/mL for 3 days). 100 nM 4-hydroxytamoxifen (4OHT) (Sigma, H7904), were used for IMR90 ER:HRAS$^{G12V}$ induction. Cells were regularly tested for mycoplasma contamination and always found to be negative.

### Vectors
The following retroviral vectors were used: pLNCX2 (Clontech) for ER:HRAS$^{G12V}$ (Addgene 67844)[15], sh-p53 (target sequence 5′-CACTA-CAACTACATGTGTA-3′)[36], pSuper for sh-CEBPB (gift from D. Peeper)[35], and MSCV-puro for miR30 sh-HMGA1 (target sequence 5′-CGCCTGGGATCTGAGTACATAT-3′)[13]. The sh-HMGA1 has been extensively utilised, although its off-targets cannot be completely excluded[47–49].

### Generation of HMGA1-deficient H1299 cells
Following CRISPR guides (Phosphorothionate-modified sgRNA) were designed against Exons 3 and 4 of HMGA1 (ENSG00000137309): Sg199_Exon 3: 5′-GCTGCTTGCGCGGCCTGCCC-3′ (PAM: CGG) and Sg202_Exon 4: 5′-CTTAGGTGTTGGCACTTCGC-3′ (PAM: TGG). H1299 cells were electroporated with 5 μg TrueCut spCas9 protein V2 (Invitrogen A36498) and 100pmol guide RNA (Synthego), using program EW-127 and SF nucleofector solution. A cell pellet was taken on days 3 and 10 post electroporation and genomic DNA was extracted from each pool (Qiagen, DNeasy blood and tissue kit, 69506). To genotype them, Exons 3 and 4 of HMGA1 were amplified by PCR (Phusion High-Fidelity DNA polymerase, NEB, cat#M0530S) and subjected to Sanger sequencing, and analysed using Synthego ICE web tool (https://ice.synthego.com) to calculate the percent editing in a pool over time. We used the following primers for exon 3,

 forward: 5′-AAGCTGATTAGGGCCAACGG-3′ and reverse: 5′-CATCTAAAGAGCCAGGGGCAAT-3′, and for exon 4, forward: 5′-CTAGGGGTCTTCTGGGCTCT-3′ and reverse: 5′-CAGGGTGACCCAACACTCTC-3′.

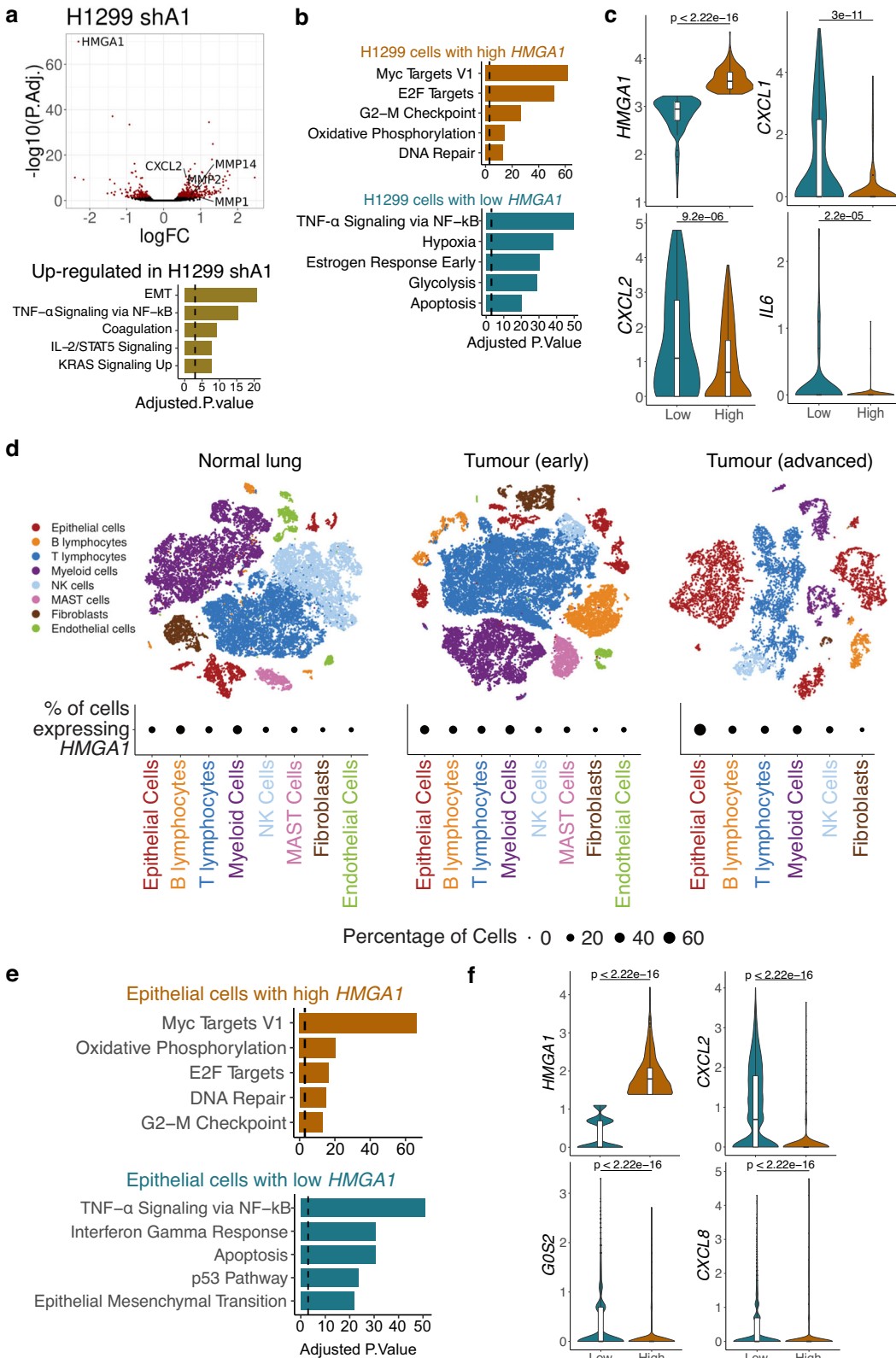

The eight pseudogenes for HMGA1 were also amplified using primer forward: 5′-GAGCTCGAAGTCCAGCCAG-3′ and reverse: 5′-TGGCAG-CACCCTTGTTTTTG-3′. We identified a knockout efficiency between 65% and 90% by 10 days post electroporation. Cas9 electroporated control pools (no sgRNA) were derived for H1299 cells. Independently generated pools (edited) were derived. To isolate single clones from the genetically edited pools, we used cellenONE X1.

## Protein expression by immunoblotting and immunofluorescence

Immunofluorescence and immunoblotting on SDS–PAGE on gels of various concentrations, were performed as described previously[13]. Briefly, for immunoblotting, the protein was transferred to PVDF membranes (Immobilon, Millipore), which were subsequently blocked for 1 hour at room temperature in a 5% milk solution in TBS-Tween 0.1%

**Fig. 6 | The effect of HMGA1 on the transcriptome of lung adenocarcinoma. a** Expression changes in H1299 cells in response to shHMGA1, highlighting differentially expressed genes (top) and gene enrichment analysis against the MSigDB Hallmarks of the genes up-regulated by shHMGA1 in H1299 cells (bottom). Expression P-values derived from edgeR differential expression testing and adjusted for multiple testing using Benjamini-Hochberg correction. **b** Gene enrichment analysis of the markers of H1299 cells with high and low HMGA1 expression from a single-cell expression dataset of H1299 cells (see Methods). **c** Distribution of the expression at single-cell level of the representative markers of the same H1299 cells from **b** with low (n = 131) and high (n = 187) expression of HMGA1, respectively.

**d** Cell populations of normal lung, early and advanced lung adenocarcinoma and the percentage of cells expressing HMGA1. **e** Top enrichment results (MSigDB Hallmarks) for the gene markers of epithelial cells in the advanced tumour with high (top) and low (bottom) HMGA1 expression. **f** Expression distribution of representative genes for the same cells from **e**, with high (n = 861) and low (n = 3084) HMGA1 expression. **a**, **b**, **e** Gene enrichment *P* values calculated with the EnrichR software using Fischer's exact test and adjusted for multiple testing with Benjamini-Hochberg. **c**, **f** *P* values derived from two-sided Wilcoxon testing. Box plot centre line represents the median, the bounds correspond to the 0.25 and 0.75 quantiles, the whiskers represent the 0.1 and 0.9 quantiles.

before incubating with the primary antibody at 4 °C overnight. An appropriate HRP-conjugated secondary antibody was incubated at room temperature for 1 hour. Immunoblots were visualized with chemiluminescence reagents (Sigma, RPN2106). For immunofluorescence, cells were plated onto #1.5 glass coverslips the day before fixation to achieve approximately 60% confluence. Cells were fixed in 4% (v/v) PFA and permeabilised with 0.2% (v/v) Triton X-100 in PBS with DAPI. Coverslips were mounted onto Superfrost Plus slides (ThermoFisher, 4951) with Vectashield Antifade mounting medium (Vector Laboratories Ltd., H-1000). The following primary antibodies were used for immunoblotting: anti-β-actin (mouse monoclonal, Sigma, A5441 (AC-15), 1:2500); anti-HRAS (mouse monoclonal, Santa Cruz, sc-29 (F235), 1:1000); anti-HMGA1 (rabbit polyclonal, Active motif, Cat #39615, 1:2000); anti-IL6 (mouse monoclonal, R&D Systems, MAB2061, 1:500); anti-IL8 (mouse monoclonal, R&D Systems, MAB208, 1:500); anti-Cyclin A (mouse monoclonal, Sigma, C4710 (CY-A1), 1:1000); anti-C/EBPβ (rabbit polyclonal, Santa Cruz, sc-150 (C-19), 1:500); anti-p53 (mouse monoclonal, Santa Cruz, sc-126 (DO-1), 1:500): anti-CDKN1A p21 (mouse monoclonal, Santa Cruz, sc-6246 (F-5), 1:500). The following primary antibodies were used for immunofluorescence: anti-H3K9me3 (mouse monoclonal, Hiroshi Kimura Laboratory, clone CMA318, 4 μg/mL), anti-HMGA1 (rabbit polyclonal, Cold Spring Harbor Labs, CS2637, 1:1000), anti-CDKN2A p16 (mouse monoclonal, BD Biosciences 554079, clone G175-1239, 1:500), and anti-IL8 (mouse monoclonal, R&D Systems, MAB208, 1:500). Cells were counter-stained with DAPI with secondary antibodies (Alexa Fluor 488, 555, 647). Fluorescence images were obtained with Leica Stellaris 8 Confocal Fluorescence Microscope. Source data is provided in the Source Data file for the quantified immunofluorescence values and uncropped blots.

## ChIP-seq

Chromatin immunoprecipitation (ChIP) was performed as previously described[50] for the following antibody: anti-HMGA1 (abcam ab192153, lot# GR3212620-2, 10 μg/50 M cells), with some modifications. Cells were fixed with 1% formaldehyde at room temperature for 10 min or at 4 °C for 1 hour. Briefly, cells were sequentially lysed in LB1 Buffer (50 mM HEPES-KOH [pH7.5], 140 mM NaCl, 1 mM EDTA, 10% glycerol, 0.5% IGEPAL CA-630 and 0.25% Triton X100) followed by LB2 Buffer (10 mM Tris-HCl [pH8.0], 200 mM NaCl, 1 mM EDTA and 0.5 mM EGTA), with occasionally mixing and incubating on ice for 10 min and 5 min, respectively. The spin-down pellet was washed once with LB3 Buffer (10 mM Tris-HCl [pH8.0], 100 mM NaCl, 1 mM EDTA, 0.5 mM EGTA, 0.1% sodium deoxycholate and 0.5% N-lauroylsarcosine). All buffers were supplemented with proteinase inhibitor cocktail (cOmplete ULTRA, EDTA-free, Roche 05892791001) and 1 mM Phenylmethanesulfonyl fluoride (Merck, 78830). The lysate in LB3 was sonicated using BioRuptor Pico (Diagenode) with an ON cycle of 30 seconds for >24 cycles and OFF intervals of 30 seconds, resulting in chromatin DNA fragments with a median size of 150 bp. The lysate was cleared by centrifugation at maximum speed (>20,000 × *g*) for 10 min. Triton X-100 was added to a final concentration of 1%, and the cleared

lysate was then utilized for immunoprecipitation. Antibodies were preincubated with Dynabeads M-280 Sheep Anti-Rabbit IgG (Invitrogen, 11204) for >4 hours at 4 °C. Spike-in chromatin (Active motif 53083, lot# 06420010, 100 ng/sample) and spike-in antibody (Active motif 61686, lot# 0642008, 4 μg/sample) were added in immunoprecipitation for normalisation. Immunoprecipitation was performed at 4 °C overnight (>16 h) with rotation of the sample tubes. The beads were washed 6 times with RIPA Buffer (50 mM HEPES-KOH [pH7.5], 500 mM LiCl, 1 mM EDTA and 1% IGEPAL CA-630, 0.7% sodium deoxycholate) and once with TBS buffer (20 mM Tris-HCl [pH 7.6], 150 mM NaCl). The immunoprecipitated DNA was eluted in ChIP Elution Buffer (50 mM Tris-HCl [pH 8.0], 10 mM EDTA and 1% SDS). Both input and immunoprecipitated DNA were incubated with RNase A (Invitrogen, AM2271) at 65 °C overnight. The DNA was then purified with Proteinase K (Invitrogen, 25530049) by incubating at 55 °C for 2 h, followed by purification using the MinElute PCR Purification Kit (Qiagen, 28004). Libraries were prepared using the NEBNext Ultra II DNA Library Prep Kit for Illumina (New England Biolabs, E7645L) according to the manufacturer's instructions except that size selection was performed after PCR amplification using AMPure XP beads (Beckman Coulter, A63881). Samples were sequenced paired-end using 50 bp reads on the Illumina platforms. Spike-in chromatin (Drosophila chromatin) and spike-in antibody (Drosophila-specific histone variant, H2Av) from Active motif was added for the HMGA1 ChIP-seq. We confirmed that the scaling factors calculated based on the spike-in reads were similar between samples.

## RNA-seq

RNA was purified using the QIAGEN RNeasy plus kit according to the manufacturer's instructions. The quality was checked using the Agilent High Sensitivity RNA Bioanalyser or ScreenTape System. Libraries corresponding to biological replicates of each condition were prepared using TruSeq stranded mRNA or Illumina Stranded mRNA Prep Kit (Illumina), according to the manufacturer's instructions, and sequenced using the HiSeq-4000 or the Novaseq-6000 platforms (Illumina).

## scRNA-seq

Two replicates of IMR90 growing cells expressing vector control and OIS IMR90 cells expressing shHMGA1 or vector control were individually labelled with 1 μg of Biolegend TotalSeq™-B Cell Hashing antibodies diluted in Cell Staining Buffer (PBS with 3%FBS) for 30 min at 4 °C, then washed 3 times with Cell Staining buffer (centrifuge 2 min 350 g at 4 °C). Cells were resuspended to a concentration of 800 cells/μL for single-cell encapsulation using the Chromium Single Cell B Chip Kit (10x Genomics Cat # PN-1000073), followed by library prep using the Chromium Single Cell 3' GEM Library & Gel Bead Kit v3 (10x Genomics Cat # PN-1000075) for the gene expression library and the Chromium Single Cell 3' Feature Barcode Library Kit (10x Genomics Cat # PN-1000079) for the hashtag-oligo library. Both libraries were then pooled for paired-end sequencing on the NovaSeq 6000 platform (Illumina).

## Hi-C

Hi-C was performed in IMR90 cells in the Grow and OIS conditions with shA1 (knock-down of HMGA1) as described before[51], matching our previously published Hi-C data for IMR90 cells[8]. Briefly, 50 million cells were used for each sample, and digestion was performed using *HindIII*. 50 million cells were fixed in 2% formaldehyde / DMEM for 10 min at room temperature and quenched with 0.125 M glycine. Cells were permeabilised for 30 min in 50 mL of permeabilization buffer (10 mM Tris-HCL pH 8, 10 mM NaCl, 0.2% IGEPAL CA-630, cComplete EDTA-free protease inhibitor cocktail (Roche 118735800001)). Cells were centrifuged at 600 × *g* for 5 min a 4 °C and suspended in 358 μL NEB-uffer 2 (NEB, B7002S) per 5 million cells (aliquoted). To remove poorly fixed protein, 11 μL of 10% SDS was added to each aliquot and incubated at 37 °C with shaking at 950 RPM for 60 min. 75 μL of 10% Triton X-100 was added and cells were incubated at 37 °C with shaking at 950 RPM for 60 min to quench. Chromatin was digested using 1500 U of *HindIII* per aliquot (NEB, R0104T) before overnight incubation at 37 °C with shaking at 950 RPM. Restriction sites were filled with Klenow (NEB, M0210L) using 10 μM Biotin-14-dATP, dCTP, dGTP and dTTP for 60 min at 37 °C. In nucleus blunt end ligation was performed using T4 DNA ligase (NEB, M0202S) in an 8 mL reaction volume including 82 μL of 10 mg/mL BSA (NEB, B9001S) for 4 hours at 16 °C. Cross-links were reversed using 60 μL of 10 mg/mL proteinase-K (Roche, 03115879001) per aliquot for 65 °C for 2 hours. Contaminating RNA was removed using 12.5 μL of RNase A (Roche, 10109142001) for 37 °C for 1 hour.

Ligated DNA was purified from the nuclei using phenol (Sigma) extraction followed by phenol/chloroform/isoamylalcohol (Sigma) extraction and ethanol precipitation. Precipitated DNA was thoroughly washed in 70% ethanol and dissolved in 25 μL TE (10 mM Tris-HCL, 1 mM EDTA, pH 8.0) per aliquot before aliquots were combined. DNA was quantified using the Qubit HS kit (as for ChIP-seq). Biotin was removed from non-ligated ends by incubating 30–40 μg of DNA with T4 DNA polymerase (NEB, M0203L) for 4 hours at 20 °C with dATP. DNA was re-purified using the phenol/chloroform/isoamylalcohol (Sigma) extraction method before ethanol precipitation, washed in 70% ethanol and dissolving the pellet in 130 μL of H2O. Sonication was performed using a Covaris (E220) instrument to generate fragments of approximately 400 bp in length. The following settings were used: Duty Factor 10%; Peak Incident Power (w): 140; Cycles per burst: 200; Time: 55 seconds. Sonicated DNA was end-repaired with T4 DNA polymerase (NEB, M0203L), DNA polynucleotide kinase (NEB, M0201L), Klenow fragment (NEB, M0210L) and dNTPs in 1× T4 DNA ligase buffer (NEB B0202S). dATP was added to the repaired ends in NEBuffer 2 with dATP and Klenow exo-(NEB M0212L) for 30 min at 37 °C. Size selection was performed using AMPure XP beads (Beckman Coulter, A63881) to select fragments between 200 and 650 bp in length. Biotinylated fragments were isolated using MyOne Streptavidin C1 Dynabeads (Life Technologies, 650.01) in Binding Buffer (5 mM Tris pH8, 0.5 mM EDTA, 1 M NaCl) for 30 min at room temperature, followed by washing in Tween Buffer (5 mM Tris, 0.5 mM EDTA, 1 M NaCl, 0.05% Tween) and No Tween Buffer (5 mM Tris, 0.5 mM EDTA, 1 M NaCl) before washing with and suspending in 1× ligation buffer (NEB, B0202S). Paired-end (PE) adapters (as provided by Illumina, but in this case ordered from Sigma and annealed inhouse) were ligated onto Hi-C ligation products still bound to MyOne beads for 2 hours at room temperature using T4 DNA ligase (NEB, M0202S), slowly rotating the samples (-10 RPM). Samples were washed with Tween Buffer and Binding Buffer before the beads were resuspended in NEBuffer 2. Hi-C DNA on beads was divided into 2.5 μL aliquots on strip tubes and amplified from the beads using 12 cycles of PCR amplification, PE PCR 1.0 and PE PCR 2.0 primers (Illumina) in 1× Phusion polymerase buffer (Phusion NEB F531) with 10 mM nucleotides using Phusion polymerase (NEB F531) in a total volume of 25 μL. PCR reactions were pooled and purified using AMPure XP beads before samples were eluted from the beads using 20 μL of TLE (10 mM Tris pH8.0, 0.1 mM EDTA). The

concentration and size distribution of libraries was determined using the KAPA library quantification kit and Agilent Bioanalyzer platforms, respectively. Hi-C samples were sequenced paired-end in individual lanes using the HiSeq 2500 and HiSeq 4000 platforms (Illumina).

## ChIP-seq analysis

Reads were aligned using Bowtie2[52] (v.2.2.4) against the hg19 genome build. No trimming was required as determined with FastQC (v.0.12, Simon Andrews). Duplicates were marked with samtools[53] (v.1.9) and blacklisted regions (hg19) were removed[54]. Samples were checked for quality control metrics and extension sizes were calculated for each sample with the R Bioconductor package ChIPQC[55] (v.1.32.2). Peaks were called with macs3 (v.3.0.0) initially with default parameters and then with the extension sizes determined. Peaks present in at least two of the replicates of a condition were combined into a consensus set for each condition. THOR[56] (v.1.0.2) was used to normalise and to perform differential binding pairwise between conditions. The THOR-normalised signal was used for visualisation with the IGV genome browser[57]. HMGA1-dense regions were calculated by counting the number of peaks in rolling windows consisting of 100 kb bins with a step of 5 kb and then computing the inflection point of these values beyond which a region was classified as 'dense'. ChIP-seq heatmaps were plotted using deepTools[58] (v.3.1.0). Peak annotation was performed with the R Bioconductor package annotatR[59] (v.1.22.0).

## Bulk RNA-seq analysis

Libraries were aligned using STAR[60] (v.2.7.10) against the hg19 genome and transcriptome (GENCODE19) reference. We used n = 8 replicates per condition. Reads were counted over genes using the GENCODE19 annotation and the featureCounts functionality from the subread package[61] (v.1.6.2). Differential expression analysis was performed pairwise using the R Bioconductor package edgeR[62] (v.3.40.2). Gene set enrichment was performed using the R interface of EnrichR[63], the enrichR package (v.3.2).

## scRNA-seq analysis

Single-cell RNA-seq libraries were processed using the CellRanger pipeline (10x Genomics)[64] against the hg19 reference transcriptome. Read counts were further analysed using the R Bioconductor package Seurat[65] (v.4.1.0). Hashtag demultiplexing was performed with HTO-Demux from Seurat and doublet identification was performed with DoubletFinder using the hashtag read counts. After selecting singlets, low-quality cells were filtered out (mitochondrial content above 5%). We used the edgeR Bioconductor package[62] (v.3.40.2) for pseudobulk analysis. The Milo R Bioconductor package[32] (v.1.4.0) was used for differential analysis testing on cell neighbourhoods. Gene set scores were calculated using UCell[66] (v.2.2.0).

## Hi-C analysis

Libraries were aligned using HiC-Pro[67] (v.3.1.0) against the hg19 genome. Experimental artefacts and duplicates were removed prior to counting reads in matrices corresponding to different resolutions. Agreement between replicates was tested with HiCRep[27] (v.1.12.2), as well as by principal component analysis of the normalised and filtered count matrices at several resolutions. A/B compartment score was determined as before[8], by performing PCA on distance-corrected ICE-normalized Hi-C matrices at 200 kb resolution using the R Bio-conductor package diffHic[68] (v.1.28.0). Differential interaction analysis was performed using the same package (diffHic). Network analysis of the differential interactions was performed using the igraph R package (v. 1.3.5, CRAN Csardi & Nepusz, 2006). The ggraph R package (v.2.2.0, Pedersen 2022 CRAN) was used for plotting the chromosomal networks. ICE-normalised and distance-corrected matrices were used for visualisation using HiCvizR[8] (v.1.0). Briefly, coordinates matching both linear epigenetic tracks as well as Hi-C matrices in triangular shape are

calculated and plotted so that the colour scale matches the normalised contact frequency.

## Image analysis

Immunofluorescence signal quantification was performed with the StarDist[69] (v.0.8.3 nuclei segmentation) and the scikit-image[70] (v.0.19.3 image processing) Python libraries. HMGA1 average signal was calculated for each labelled cell nucleus and the IL-8 average signal was calculated over an area around the nuclear boundary.

## Statistics and reproducibility

Genomics data and quantified immunofluorescence measurements were statistically analysed using R (v.4.2.2). Immunoblotting and immunofluorescence experiments were repeated twice or three times for each condition. No data were excluded from the analyses. Analyses of distributions of immunofluorescence intensity values, gene expression log-fold changes, enhancer connectivity, and genomic features were performed pairwise between pairs of conditions with P-values derived using two-sided Wilcoxon tests and visualised using the median and the 0.1, 0.25, 0.75, and 0.9 quantiles. Gene enrichment analysis was performed using Fischer's exact test and the Benjamini–Hochberg multiple testing correction.

## Reporting summary

Further information on research design is available in the Nature Portfolio Reporting Summary linked to this article.

## Data availability

Source data are provided as a Source Data file. All sequencing data generated during this study are available from the Gene Expression Omnibus (GEO) under the accession ID GSE245808. It contains bulk RNA-seq, scRNA-seq, Hi-C, and ChIP-seq data. We also used the following datasets which we previously published and deposited on GEO: Hi-C IMR90, CTCF and RAD21 ChIP-seq–GSE135093 [8], Lamin B1 ChIP-seq–GSE49341 [22], histone marks and ATAC-seq–GSE103590 [16] and GSE38442 [71], p53 ChIP-seq–GSE53491 [36], and C/EBPβ ChIP-seq–GSE180358 [17]. Other public datasets (single-cell RNA-seq) we used are GSE131907 [40] (lung adenocarcinoma) and GSE183590 [39] (H1299 cell line). Source data are provided with this paper.

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

## Acknowledgements

We thank all members of the Narita laboratory for editing and suggestions, particularly Dr. Andrew Young and Aleksandra Janowska, and the staff of the Cancer Research UK Cambridge Institute (CRUK-CI) core facilities (Genomics, Microscopy, Genome Editing, Bioinformatics, and RICS). This work was supported by the CRUK-CI core grant (C9545/A29580) to the M.N. laboratory. M.N. was also supported by BBSRC (BB/S013466/1 and BB/T013486/1), Diabetes UK via BIRAX, and the British Council (65BX18MNIB). S.S. was supported by the Babraham Institute's Epigenetics Institute Strategic Programme (BBS/E/B/000C0421), a UKRI MRC Rutherford Fund Fellowship (MR/T016787/1), and a Career Progression Fellowship from the Babraham Institute. I.O. was also supported by a Cancer Research UK Pioneer Award (C63389/A30462). Y.O. was supported by JSPS KAKENHI (JP23H00372, JP24H02323), AMED BINDS (JP22ama121017jO001), MEXT Promotion of Development of a Joint Usage/Research System Project: Pan-Omics DDRIC, MRCI for High Depth Omics, CURE: JPMXP1323015486 for MIB and RIIT in Kyushu Univ.

## Author contributions

I.O. and M.N. conceived the study with input from M.A.-K. and A.J.P. Experiments and data analysis were conducted by M.A.-K., A.J.P., S.S., T.H., Masako N., and Y.O. Computational analyses were conducted by I.O. and M.A.-K. Hi-C experiments were performed by A.J.P. and S.S.,

under P.F.'s supervision. A.J.P. and T.H. performed the ChIP-seq experiments with help from H.K. The RNA-seq and scRNA-seq experiments were performed by A.J.P., M.A.-K., and Masako. N., with help from Y.O. The establishment of the HMGA1 knock-out H1299 cell lines was led by M.A.-K. Histone antibodies were provided by H.K. The manuscript was written by I.O. and M.N. with input from the other authors.

## Competing interests

A.J.P. is an employee of Altos Labs. P.F. and S.S. are co-founders and shareholders of Enhanc3D Genomics Ltd. All other authors declare no competing interests.

## Additional information

[1]Cancer Research UK Cambridge Institute, Li Ka Shing Centre, University of Cambridge, Cambridge, UK. [2]Epigenetics Programme, The Babraham Institute, Cambridge, UK. [3]Nuclear Dynamics Programme, The Babraham Institute, Babraham Research Campus, Cambridge, UK. [4]Division of Transcriptomics, Medical Institute of Bioregulation, Kyushu University, 3-1-1 Maidashi, Higashi, Fukuoka 812-0054, Japan. [5]Cell Biology Center, Institute of Innovative Research, Tokyo Institute of Technology, Yokohama, Japan. [6]Tokyo Tech World Research Hub Initiative (WRHI), Institute of Innovative Research, Tokyo Institute of Technology, Yokohama, Japan. [7]Present address: Division of Tumor Biology and Immunology, The Netherlands Cancer Institute—Oncode In stitute, The Netherlands Cancer Institute, Amsterdam, The Netherlands. [8]Present address: Altos Labs Cambridge Institute, Portway Building, Granta Park, Cambridge, UK. [9]Present address: Enhanc3D Genomics Ltd, Cambridge, UK. [10]These authors contributed equally: Ioana Olan, Masami Ando-Kuri, Aled J. Parry.
✉e-mail: Masashi.Narita@cruk.cam.ac.uk

