## [Peer Review File · Nature Communications]

REVIEWER COMMENTS

Reviewer #1 (Remarks to the Author):

HMGA1 plays a key role in chromatin re-organization in cellular senescence, contributing to SAHF formation, senescence stability, and gene expression. The authors nicely integrate different high throughput approaches to find that HMGA1-mediates chromatin environment and chromatin compartmentalization. They suggest that these functions of HMGA1 mediate the regulation of gene expression and contribute to the heterogeneous nature of senescence. This interesting and well-performed work addresses the fundamental role of HMGA1-mediated chromatin compartmentalization and gene regulation. There are some comments to be addressed, and they are listed below.

Main questions:

1. Does HMGA1 drive a unique 3D chromatin organization only in OIS? What is its role, if any, in #D chromatin organization in growing cells? In damage-induced senescent cells?
2. How specific are the observed effects of OIS induced by high levels of Ras in IMR90 cells? Could similar effects be observed in other primary mouse or human cells induced to senescence with another oncogene?

Specific questions:

1. In Figure 1 authors show that HMGA1 depletion suppressed SAHF formation with little impact on senescence arrest. Could authors show the protein levels of p16 in OIS-shA1, known to be elevated in OIS, as this is missing in Fig. 1a? Showing expression of cell cycle related gene signatures from the RNA-seq might be helpful.
2. It is very interesting that HMGA1 has a modulatory effect on the expression of inflammatory genes acting as a transcriptional buffer for SASP (Fig 1b,c). What could be the physiological relevance of this process in limiting the inflammatory secretome in OIS by HMGA1?

3. The authors suggest that HMGA1 acts as a transcriptional buffer for SASP and ECM genes. However, from this study one might conclude that the effects are the result of overall chromatin rearrangement and are not specific to SASP or ECM. This should be further clarified.

4. Cells with depletion of HMGA1 (OIS-shA1) produce an extensive amount of metalloproteins and proinflammatory cytokines eg IL-8 (Fig.1). Energy-wise it is very expensive cost for cells. Does HMGA1 is known to modulate genes of metabolic pathways?

5. Authors report that HMGA1 binds promoters clustered in type-I interferon loci (e.g., IFNAs and IFNB1) however IFNA was not expressed in OIS IMR90 cells at the day 6 of senescence. The authors suggested that IFNA and IFNB1 are induced only in 'late senescent'. Is the binding of HMGA1 to chromatin and compartmentalization of chromatin dynamic change throughout senescence? The authors previously described the expression changes in late vs early senescence. Are these changes consistent with constitutive HMGA1 binding and its role in senescence stability?

6. HMGA1 plays a pivotal role in SAFH formation and gene regulation in senescence. HMGA1 incorporates genes in heterochromatic cores contributing to their down-regulation but also excludes genes from cores contributing to their upregulation in OIS (Fig.3). Cellular senescence (and its gene expression signature) can have physiological, but also pathological roles depending on the biological context. Could authors speculate if it would be possible to modulate the binding of HMGA1 to chromatin to regulate gene expression in early/late senescence for therapeutic intervention?

7. The authors mention that senescence is heterogeneous and suggest that HMGA1 contributes to the heterogeneity. What are the consequences of this at the cell population level? Are there cells with different levels of HMGA1 and do these cells present features that are closer to the early or late senescence depending on HMGA1 expression?

8. Overall, it could be helpful if authors would better explain the connections of their data and conclusions to physiological contexts where senescence plays a role.

Reviewer #2 (Remarks to the Author):

This manuscript by Olan, Kuri and colleagues uses the well established OIS system in IMR90 to address the on-chromatin role of HMGA1, a nuclear factor long-known for its contribution to OIS phenotypes, in chromatin reorganization and senescence commitment. The authors combine 3D genomics with a nice graph-theory-based approach to draw dynamic chromatin networks associated with HMGA1 bound to its nodes. In my opinion, such a study into HMGA1 in this context has long been awaited and sheds new light on the generally underappreciated roles of HMGA in senescence and homeostasis. Overall, the study is well conducted and well controlled, and the generalization of the findings in a lung cancer system renders it more accessible to a larger readership, with the role of HMGA now taking new shape also in the context of malignancy. I offer below some suggestions that could further improve the message of this otherwise very good manuscript:

- I understand why so many HMGA1 peaks emerge in the ChIP-seq data, but have three suggestions. First, what if "broad peaks" are called since the signal looks almost continuous along loci? Second, how do these ChIP-seq profiles compare to those obtained earlier for HMGA by the Schubeler lab? Third, it is unclear how HMGA1-dense regions are called and how this compares to "broad peak calling"? Please clarify this in the main text for readers.

- The proposition that HMGA1-dense regions in the lamina are more prone to Lamin B1 loss and thus likely contributors to SAHF formation is very interesting. Is there a way to substantiate this further? The authors have quite a bit of understanding of SAHF that should render this possible? For example, can some exemplary regions be shown/validated? This also refers to the graph-based analysis in Fig. 2.

- The localization of HMGA1 to a subset of enhancers is very interesting. However, unless I missed it, we are not shown how HMGA1-bound enhancers in proliferating cells behave in OIS ones and what genes they are connected to?

- We find out very little about the Hi-C data quality, read depth and resolution in the main text, which would be useful for assessing how cleanly HMGA1-relevant connections are portrayed in this data and at which resolution. The connections drawn are really interesting, but resolution and dynamic range of signal in the Hi-C data need to be well presented. More generally, I would have preferred more technical details from the Methods to make it out of the main text to improve clarity.

- I find the trend lines in Fig. 2b to be not so convincing. It appears as if the change in curvature is mostly due to few interactions and not in the more dense part of the data cloud. In addition, I would not fully subscribe to the use of "homo-" and "heterotypic" interactions used here, as both regions more or less bind HMGA1, so are likely homotypic. Unless the authors can provide additional analysis (e.g., histone mark content) of these interacting regions, this panel should be either removed altogether or heavily revisited as regards its interpretation.

- Not sure if "saddle plots" would make the message of Fig. 2C more apparent?

- The scRNA-seq data are a really nice addition, but their UMAP representation is now generally thought to be rather unrepresentative of data clustering. Would tSNE maps separate as nicely too? This is merely a suggestion. Moreover, I would like to see a statement in the main text on the

apparent mixing of the OIS and OIS-shA1 populations, suggesting that HMGA1 regulation does not strongly change the senescent profile of the cells?

- I would urge the authors to add the lung cancer data as a main figure given how well it rounds up their manuscript.

- Finally, I was wondering if HMGA1 overexpression in proliferating cells is doable and if this can impose some of the OIS effects to "young" cells? I know that HMGB1/2 overexpression is quite deleterious to IMR90 even in the short-term, but is this so for HMGA1 too? This would be a way to assess if HMGA1 levels alone are a driver of some of these changes? Currently, the only unclear point of this work lies with the fact that HMGA1 is present in both cell states, but in OIS is somehow redirected such that it can control a considerable part of the OIS program.

Reviewer #3 (Remarks to the Author):

General points

This is a well-executed study that digs into the biological roles of HMGA1 as a chromatin organizer, and how these interactions are modulated in senescence, and in turn, how this impacts the gene expression changes that are observed in senescence. The study exploits the elegant and well-characterized model of OIS, in which a stably integrated Ha-RasER protein is activated with tamoxifen. Superimposed on this is the downregulation of HMGA1, which is achieved by a transduction of a well-characterized shRNA against HMGA1 in a miR30 design. A diverse array of genomic methods are applied to analyze this biological system. The experiments are well executed and documented, and the analysis is comprehensive. The conclusions are largely consistent with prior information, but the data provide a much sharper focus and clarity. There are many interesting features that will be of interest to those in this field. The further elucidation of the chromatin architectural roles of HMGA1, especially how it shapes heterochromatin, will be of wide interest. The one exception is that the data are not consistent with a role of HMGA1 as a direct regulator of gene expression. I view this also as a valuable contribution to the field.

I only have three specific points:

1) In a general sense, using only one shRNA sequence, controlled by empty vector (not scrambled sequence) is not rigorous enough to exclude off target effects. I understand that this specific shRNA has been the subject of several papers and that these issues have been addressed in the past. However, for completeness of this paper, this information should be summarized so that the reader can be reassured that off-target effects have been rigorously addressed. In my opinion a few sentences in the Methods section would suffice.

2) Given the wide-spread nature of HMGA1's effect on gene expression, some of the effects observed could be secondary (indirect) in nature (HMGA1 affecting the expression of some gene which in turn influences the particular result being measured). The good correlation of HMGA1 binding with many of the gene expression (and other) changes indicate that many of the effects are direct. However, indirect effects cannot be ruled, and I feel this should be briefly mentioned in the Discussion.

3) I found Fig. 4F confusing. Maybe the authors can try to redraw this? At minimum, I would like a more detailed explanation in the legend on how to interpret the figure.

We thank all three reviewers for the highly salient comments and suggestions, which we have addressed below. We are grateful for the opportunity to strengthen some of the points made in the manuscript and add more details, which were perhaps understated in the original version.

Reviewer #1:

HMGA1 plays a key role in chromatin re-organization in cellular senescence, contributing to SAHF formation, senescence stability, and gene expression. The authors nicely integrate different high throughput approaches to find that HMGA1 mediates chromatin environment and chromatin compartmentalization. They suggest that these functions of HMGA1 mediate the regulation of gene expression and contribute to the heterogeneous nature of senescence. This interesting and well-performed work addresses the fundamental role of HMGA1-mediated chromatin compartmentalization and gene regulation. There are some comments to be addressed, and they are listed below.

Main questions:

1. Does HMGA1 drive a unique 3D chromatin organization only in OIS? What is its role, if any, in 3D chromatin organization in growing cells? In damage-induced senescent cells?

HMGA1 affects the 3D organisation of cells beyond OIS as can be seen in the normal fibroblasts with sh-HMGA1 (Grow-shA1), where compartmentalisation is similarly reduced (original Extended Data Fig. 3g, now Supplementary Fig. 3i). Long-range (heterochromatic) interactions are also generally reduced, albeit weaker, when HMGA1 is knocked down in normal fibroblasts (Fig. 2a). These results suggest that a direct role for HMGA1 in the chromatin organisation seen in OIS cells is conserved in normal fibroblasts. We have reinforced this point in the revised manuscript. In DNA damage-induced senescence, upregulation of HMGA1 and SAHF formation are typically modest (Narita et al. *Cell* 2006 PMID 16901784, Tomimatsu et al. *Nat Aging* 2022, PMID 37118356), compared to OIS. Thus, we suspect that the HMGA1-dependency in 3D chromatin organisation is also milder, although this will need to be directly tested.

2. How specific are the observed effects of OIS induced by high levels of Ras in IMR90 cells? Could similar effects be observed in other primary mouse or human cells induced to senescence with another oncogene?

The OIS phenotype is largely conserved across various primary human fibroblasts. In particular, IMR90, WI38, and BJ cells have been employed in numerous studies as an OIS model (Serrano et al. *Cell* 1997 PMID 9054499, Narita et al. 2003 *Cell* PMID 12809602, Beausejour et al. *EMBO J.* 2003 PMID 12912919). Generally, IMR90 and WI-38 cells (human embryonic lung fibroblasts) represent a highly stable senescence model. We and others have shown that this stability of senescence arrest is well correlated with p16, SAHF formation and HMGA1 (Narita et al. *Cell* 2006 PMID 16901784, Beausejour et al. *EMBO J.* 2003 12912919). BJ cells (human neonatal skin fibroblasts) exhibit less stable senescence arrest, partly due to a lower level of p16 (Beausejour et al. *EMBO J.* 2003 PMID 12912919).

Notably, while HMGA1 is a direct structural component of SAHFs, p16 (and its downstream effector, RB) **functionally** contributes to SAHF formation. Indeed, OIS BJ cells show less prominent SAHFs (Narita et al. *Cell* 2006 PMID 16901784). These studies collectively support the critical role of 3D chromatin organisation towards the strength of the senescent phenotype and its heterogeneity.

While RAS is the prototype of a senescence-inducing oncogene, other oncogenes in this signalling pathway, such as MEK and RAF, can also induce senescence with prominent SAHFs (e.g., Narita et al. *Cell* 2003 PMID 12809602). OIS WI38 cells, for example, also exhibit strong compartmentalisation and increased long-range interaction in Hi-C (Sati et al. *Mol Cell* 2020 PMID 32220303). Typically, excessive mitogenic activities (e.g., RAS/MAPK) can induce senescence (which is resistant to apoptosis) and other 'immortalising (i.e., inhibiting senescence)' oncogenes like MYC promote apoptosis. This is the basis for the cooperative effects of these two distinct types of oncogenes in malignant transformation, although recent studies suggest that MYC may also facilitate senescence (Lopes-Paciencia et al. *Cell Rep.* 2024 PMID 38568812).

Mouse embryonic fibroblasts (MEFs) can be seen as the mouse counterpart of human primary fibroblasts. However, they show substantial differences in terms of senescence signalling. For example, MEFs can be immortalised and transformed more readily than human fibroblasts. It is well-established that the state of 'senescence' in MEFs is highly unstable: the classical **3T3 protocol** (cells are split every 3 days with a fixed seeding density of $3 \times 10^5/6$ cm dish) allows for obtaining immortalised MEFs simply by regular passages under the atmospheric oxygen concentration (~21%). In addition, unlike human fibroblasts, Campisi's lab has shown in their seminal work in 2003 (Parrinello et al. *Nat Cell Biol.* 2003 PMID 12855956) that MEFs are immortal under physiological oxygen concentration (3-5% O₂) (we routinely use 5% O₂ incubators for all primary cell lines). Therefore, MEFs are generally useful for studying **senescence bypass/transformation**, to which human fibroblasts are highly resistant. In addition, mouse cells have long centromeres and telomeres (somatic cells express telomerase), the former resulting in the formation of **chromocenters** (constitutive heterochromatic foci), making it challenging to assess SAHF formation. Indeed, the Adam group has suggested that MEFs do not form robust SAHFs (Kennedy et al. *Cell Div.* 2010 PMID 20569479). Therefore, this study focuses on human cells to investigate robust senescence. Nevertheless, we hope to extend our study to rodents in the future, using *in vivo* models to investigate in particular early tumorigenesis.

Specific questions:

1. In Figure 1 authors show that HMGA1 depletion suppressed SAHF formation with little impact on senescence arrest. Could authors show the protein levels of p16 in OIS-shA1, known to be elevated in OIS, as this is missing in Fig. 1a? Showing expression of cell cycle related gene signatures from the RNA-seq might be helpful.

We have included the protein levels of p16 in the same lysates in Fig. 1a (Reviewer Fig. R1a). Consistent with our previous study, HMGA1 knockdown shows little impact on the p16 level (Narita et al. *Cell* 2006 PMID 16901784).

Reviewer Fig. R1. a, Protein levels (WB) in Growing IMR90 (d0), OIS (d6), and OIS with HMGA1 knockdown cells (d6 shA1) including p16/CDKN2A; **b**, expression log-fold changes in the OIS vs Grow and OIS-shA1 vs OIS pairwise comparisons of E2F-target (cell-cycle) genes (MSigDb gene set). These figure panels are included in the revised manuscript (Fig. 1a and Supplementary Fig. 1g).

We have also included the expression of cell cycle-related genes in the new Supplementary Fig. 1g (Reviewer Fig. R1b). As an example, we used the E2F targets gene list from the Molecular Signatures Database (MSigDb, <https://www.gsea-msigdb.org/gsea/msigdb/>), in which, the majority of genes are unaltered with HMGA1 knockdown in the OIS context. While some genes are altered, they show only a modest change with the knockdown. This supports the phenotypic observation: "little impact of senescence arrest". This additional bulk-RNA-seq data reinforces the critical message of our scRNA-seq data (original Fig. 4d,f, now Fig. 5d,f), highlighting the heterogeneity of senescence features and stability depending on HMGA1 levels. We thank the reviewer for this suggestion.

2. It is very interesting that HMGA1 has a modulatory effect on the expression of inflammatory genes acting as a transcriptional buffer for SASP (Fig 1b,c). What could be the physiological relevance of this process in limiting the inflammatory secretome in OIS by HMGA1?

We agree this is very interesting. Currently, the functional relevance of the buffer effect of HMGA1 on the SASP and other genes remains elusive. However, at a cell population level, this effect contributes to the senescence diversity, which might shape the 'senescence niche' *in vivo*. For example, p16 and SASP are both considered to be senescence hallmarks, but our scRNA-seq data

suggest that they represent distinct subpopulations within the entire OIS cells and that the HMGA1 buffer effect contributes to this senescence diversity. Such intra-population heterogeneity could also be highly relevant in the cancer context, where cellular and HMGA1 heterogeneity might contribute to finetuning the inflammatory niche (original Extended Data Fig. 8, now Fig. 6). We emphasised this point in the Discussion.

3. The authors suggest that HMGA1 acts as a transcriptional buffer for SASP and ECM genes. However, from this study one might conclude that the effects are the result of overall chromatin rearrangement and are not specific to SASP or ECM. This should be further clarified.

HMGA1 binds highly AT-rich sequences and as such, SASP and ECM genes may have a higher propensity to being affected by HMGA1 in specific ways due to the sequence compositions of their gene bodies and the genome around them. The chromatin rearrangements reflect the level of HMGA1 binding and, therefore, the sequence properties of the regions involved. We have analysed the distribution of the AT content (%) of different gene sets from the MSigDb Hallmarks collection and, interestingly, highlighted "Protein Secretion" as the most "AT" genes, whereas "Myogenesis" are the least "AT" (relative to the 55% threshold) (Reviewer Fig. R2). We have observed that pathways we identified as affected by HMGA1 have specific distributions of AT. Cell cycle genes ("E2F targets" and "G2-M Checkpoint") and SASP/ECM genes ("Inflammatory Response") have high AT content and therefore are more likely to be bound by HMGA1. This might explain why similar genes are affected by HMGA1 in cancer (H1299 cells). We clarified in the text that the sequence features (AT-content) of genes may explain the similarity at the transcriptome level of the HMGA1 effects in OIS and cancer.

We hope to extend this point further in our follow-up studies, potentially providing insights into the fundamental biological significance of AT abundance in genic (mainly intronic/non-coding) regions. We thank the reviewer for raising this point.

Reviewer Fig. R2. AT% distribution of all genes and of specific gene sets from the MSigDb Hallmarks collection (Broad Institute).

4. Cells with depletion of HMGA1 (OIS-shA1) produce an extensive amount of metalloproteins and proinflammatory cytokines eg IL-8 (Fig.1). Energy-wise it is very expensive cost for cells. Does HMGA1 is known to modulate genes of metabolic pathways?

This is an interesting point. Based on the literature search, HMGA1 has been implicated in glucose metabolism by positively regulating the FOXO1 transcription factor and its targets, such as *INSR* (Chiefari et al. *Front Endocrinol.* 2018 PMID 30034366). While our pathway enrichment analysis identified 'Glycolysis' in genes up-regulated by HMGA1 knockdown in the OIS context (Extended Data Fig. 1d, now Supplementary Fig. 1d), we failed to find alterations in the known effector genes (e.g., *INSR*, *FOXO1*, *IGFBP1*, *PCK1*). The role of HMGA1 in energy metabolism remains understudied, and further investigation in the context of senescence will be required. Also, please note that HMGA1 knockdown leads to reduced fibrogenic factors, including collagens and other secretory factors. Thus, it is important to carefully assess the assumption of the overall increase in energy demands upon HMGA1 depletion.

5. Authors report that HMGA1 binds promoters clustered in type-I interferon loci (e.g., *IFNAs* and *IFNB1*) however *IFNA* was not expressed in OIS IMR90 cells at the day 6 of senescence. The authors suggested that *IFNA* and *IFNB1* are induced only in 'late senescent'. Is the binding of HMGA1 to chromatin and compartmentalization of chromatin dynamic change throughout senescence? The authors previously described the expression changes in late vs early senescence. Are these changes consistent with constitutive HMGA1 binding and its role in senescence stability?

The study by Sedivy's group we cited (De Cecco et al. *Nature* 2019 PMID 30728521) researched long-term senescence evolution and showed activation of the interferon genes at 2-3 weeks after RAS-transduction in human fibroblasts. However, in our 4OHT-inducible OIS model, *IFNA* and *IFNB1* are not expressed. We typically do not maintain OIS cells beyond ~1 week because the small population with incomplete senescence inevitably grows out. This is most likely due to the heterogeneity highlighted in our scRNA-seq data (original Fig. 4, now Fig. 5).

The binding of HMGA1 in our cell models is mostly conserved in control and senescent fibroblasts, as well as in H1299 lung cancer cells, with a small number of fluctuations relative to the total number of peaks. For example, the interferon locus is also highly bound by HMGA1 in H1299 cells (Reviewer Fig. R3 below). It remains to be elucidated how *IFNA* and *IFNB1* are expressed in late senescence at the 3D chromatin level (De Cecco et al. *Nature* 2019 PMID 30728521). We speculate that 3D chromatin organisation, rather than HMGA1 binding itself, might be more dynamic. As we touched on in our discussion, our current hypothesis is that, in addition to the total level of HMGA1, qualitative aspects of HMGA1 (e.g., post-translational modifications, akin to histone code) may provide an additional gene regulatory layer through altering 3D chromatin organisation. This notion is supported by an early study which suggests a role for acetylation of HMGA1 Lys-64 or Lys-70 differentially modulates *IFNB1* expression (Conte et al. *Cell Death Differ.* 2017 PMID 28777374).

Reviewer Fig. R3. HMGA1 binding at the Interferon gene locus in the IMR90 (both growing and OIS) and H1299 cell lines.

6. HMGA1 plays a pivotal role in SAFH formation and gene regulation in senescence. HMGA1 incorporates genes in heterochromatic cores contributing to their down-regulation but also excludes genes from cores contributing to their upregulation in OIS (Fig.3). Cellular senescence (and its gene expression signature) can have physiological, but also pathological roles depending on the biological context. Could authors speculate if it would be possible to modulate the binding of HMGA1 to chromatin to regulate gene expression in early/late senescence for therapeutic intervention?

This is an excellent point. HMGA1 binds minor grooves at AT-stretches, and numerous minor groove binders (MGBs), which displace HMGA proteins from DNA, have been identified/developed. We previously showed that membrane-permeable MGBs displace HMGA1 and disrupt SAFHs in live OIS cells (Narita et al. *Cell* 2006 PMID [16901784](https://pubmed.ncbi.nlm.nih.gov/16901784/)). Historically, HMGA proteins are promising therapeutic targets in cancer and these minor groove binders (e.g., distamycin) are considered to be anti-cancer drugs. However, the challenges include their toxicity and limited specificity, which hinders their use in clinical practice. The toxicity is partly due to the context-dependent roles of HMGA1: while HMGA1 is a well-established oncogene and senescence effector, it has also been implicated in stemness and differentiation. Our current study highlights both general and context-dependent roles for HMGA1 in gene regulation. Even in the same biological context, depending on the chromatin environment, its contribution to gene regulation differs. Therefore, general inhibitors like MGBs may not be a promising strategy.

We speculate (see Discussion, also reviewer #1-5) that diverse HMGA1 gene regulation is achieved by different HMGA1 modifications and/or interaction partners and our future research will focus on determining which of these aspects contribute to specific patterns of binding and could be targeted for therapeutic intervention for senescence/ageing as well as cancer.

7. The authors mention that senescence is heterogeneous and suggest that HMGA1 contributes to the heterogeneity. What are the consequences of this at the cell population level? Are there cells with different levels of HMGA1 and do these cells present features that are closer to the early or late senescence depending on HMGA1 expression?

In our response to reviewer #1-2, we discussed the potential impact of HMGA1 heterogeneity on the inflammatory phenotype at a population level. In addition, our abundance neighbourhood (Milo) analysis of scRNA-seq data (original Fig. 4c,d, now Fig. 5c,d) also provide insights into the long-standing question about the stability of senescence.

We previously showed that, while HMGA1 depletion shows little impact on senescence arrest, stability of arrest is weaker (e.g., long-term colony formation efficiency is marginally increased). Our

scRNA-seq data suggest that a subset of RAS-expressing cells with low HMGA1 show incomplete arrest (cell-cycle genes are expressed, Cluster 3 in Fig. 5d). This sub-population likely escapes OIS in the long-term culture. However, when cells commit to the senescence fate, low HMGA1 contributes to a stronger SASP phenotype (Cluster 1 in Fig. 5d). Based on our Hi-C analysis, we suspect that the low-HMGA1 high-SASP OIS sub-population corresponds to less prominent chromatin compartmentalisation/SAHFs. What is the decision between these two paths remains to be elucidated. Notably, the high-HMGA1 OIS sub-population also shows high p16, critical for SAHF formation and senescence stability (Narita et al. *Cell* 2006 PMID 16901784). Thus, these cells might represent the most stable senescence (with modest SASP) (Cluster 2 in Fig. 5d).

Regarding the OIS kinetics, we previously identified an 'early phase' (day 3-4) during OIS in our 4OHT-inducible IMR90 system exhibit NOTCH signature, characterised by enhanced fibrogenic features and low inflammatory signature (Hoare et al. *Nat Cell Biol.* 2016 PMID 27525720). Interestingly, a small sub-population even at the established phase (day 6) resembles the NOTCH phase, which is over-represented in OIS compared to OIS-shA1 (Cluster 4 in Fig. 5d).

These data provide an integrative view of previously described diverse senescence phenotypes. We have emphasised this in the new legend of the original Fig. 4f, now Fig. 5f (see reviewer #3-3).

8. Overall, it could be helpful if authors would better explain the connections of their data and conclusions to physiological contexts where senescence plays a role.

This is also related to reviewer #1-2, in which we have discussed the potential relevance of HMGA1-mediated senescence diversity in shaping senescence/immune niches. Please also see our responses to reviewer #1-3 and 7 above: in individual cells, HMGA1 could contribute to the predominance of distinct senescence features, which are collectively thought to be senescence hallmarks. As this reviewer points out (reviewer #1-7), such diversity likely plays a key role at a **population** level.

In addition to the role of HMGA1 in regulating individual senescence features, we would like to emphasise its architectural role in gene regulation. Our study proposes that HMGA1 provides an **additional layer of gene regulation** through global chromatin configuration, rather than the local cis/trans arrangement (the traditional view) (Supplementary Fig. 1a). In the revised manuscript, we have clarified these points in the main text.

Reviewer #2:

This manuscript by Olan, Kuri and colleagues uses the well established OIS system in IMR90 to address the on-chromatin role of HMGA1, a nuclear factor long-known for its contribution to OIS phenotypes, in chromatin reorganization and senescence commitment. The authors combine 3D genomics with a nice graph-theory-based approach to draw dynamic chromatin networks associated with HMGA1 bound to its nodes. In my opinion, such a study into HMGA1 in this context has long been awaited and sheds new light on the generally underappreciated roles of HMGA1 in senescence and homeostasis. Overall, the study is well conducted and well controlled, and the generalization of the findings in a lung cancer system renders it more accessible to a larger readership, with the role of

HMGA1s now taking new shape also in the context of malignancy. I offer below some suggestions that could further improve the message of this otherwise very good manuscript:

- I understand why so many HMGA1 peaks emerge in the ChIP-seq data, but have three suggestions. First, what if "broad peaks" are called since the signal looks almost continuous along loci? Second, how do these ChIP-seq profiles compare to those obtained earlier for HMGA1 by the Schubeler lab? Third, it is unclear how HMGA1-dense regions are called and how this compares to "broad peak calling"? Please clarify this in the main text for readers.

We experimented with different peak callers: macs3 (Zhang et al. *Genome Biol.* 2008 PMID 18798982) calling narrow and broad peaks, as well as epic2 (Stovner et al. *Bioinformatics* 2019 PMID 30923821), which is advertised as better suited for H3K9me3 broad profiles. Overall HMGA1 peaks are narrow and more similar to H3K27ac/CTCF than to H3K27me3, as can be seen in the zoomed-in part of Fig. 1f. HMGA1 signal only appears as broad peaks similar to H3K9me3 at low resolution (e.g., whole chromosome view) due to the high density of narrow peaks in these regions. We include here a comparison between peak width distributions:

Reviewer Fig. R4. Peak width (macs3) distributions defined as the number of base pairs of the peak regions called from ChIP-seq experiments in growing IMR90 cells against HMGA1, CTCF and histone marks H3K27ac and H3K27me3.

The HMGA1-dense regions are called similarly to super-enhancers, based on binding density, as mentioned in the Methods section in the ChIP-seq analysis section. HMGA1 peak density is summarised over overlapping bins (100kb rolling windows with 5kb steps) and the inflection point of these values is calculated. Regions with a density higher than this value were considered 'HMGA1-dense' and were stitched together if they overlapped. We now included this information in the main text.

The resolution of the HMGA1 ChIP-seq data from the study by Schubeler lab (Colombo et al. *PLoS Genet.* 2017 PMID 29267285), which uses a constitutively expressed transgenic approach in mouse embryonic stem cells (mESC), is sufficient for deriving chromosome-wide trends but the signal-to-noise ratio is too low for deriving gene-level information about HMGA1 binding (see Reviewer Fig. R5a below showing chromosome 4 overall binding and Reviewer Fig. R5b - the interferon gene locus, which can be compared to the equivalent human interferon gene locus in our HMGA1 ChIP-seq data:

Reviewer Fig. R3 in our response to reviewer #1-5). While the study by the Schubeler lab used mouse cells, our data are largely consistent with their chromosome-wide profiling, such as the AT preference. We have cited this study (Colombo et al. *PLoS Genet* 2017 PMID [29267285](https://pubmed.ncbi.nlm.nih.gov/29267285/)) in this regard.

Reviewer Fig. R5. a, Chromosome 4: HMG A1 ChIP-seq in mESC and NP cells (log₂ ratio relative to mutant HMG A1); **b**, Interferon gene locus: HMG A1 ChIP-seq in mESC (mouse embryonic stem cells) and NP (neuronal progenitor) cells (log₂ ratio relative to mutant HMG A1).

- The proposition that HMG A1-dense regions in the lamina are more prone to Lamin B1 loss and thus likely contributors to SAHF formation is very interesting. Is there a way to substantiate this further? The authors have quite a bit of understanding of SAHF that should render this possible? For example, can some exemplary regions be shown/validated? This also refers to the graph-based analysis in Fig. 2.

We have attempted to validate some of the highly heterochromatic and HMG A1-dense regions using FISH, but we have been unable to obtain reliable probe signals, likely due to the highly inaccessible environment, as also previously suggested by the Adams lab (Zhang et al. *MCB* 2007 PMID [17242207](https://pubmed.ncbi.nlm.nih.gov/17242207/)).

From a network perspective, the (k-core) interaction classification reiterates this point: Lamina-associated domains (LADs) with HMG A1-dense regions tend to lose Lamin B1 binding (corresponding to 'Cores') during OIS and regions with less HMG A1 tend to be retained (corresponding to 'AltCores'). We now included an example comparing HMG A1 binding to the LAD status of the genomic regions in Grow and OIS (see below) as new Supplementary Fig. 4c. We thank the reviewer for suggesting this.

Reviewer Fig. R6 (new Supplementary Fig. 4c). Differential interactions network of chromosome 3 comparing the OIS and Grow conditions, highlighting the overlap between the bins (nodes) with: left - HMGA1 peaks (OIS), middle - Lamin B1 associated domains (LAD) in the Grow condition, and right - LAD in the OIS condition (black nodes - overlapping LAD, white nodes - no overlap with LAD).

- The localization of HMGA1 to a subset of enhancers is very interesting. However, unless I missed it, we are not shown how HMGA1-bound enhancers in proliferating cells behave in OIS ones and what genes they are connected to?

We have clarified in the text that HMGA1-dense enhancers show a large overlap between proliferating and OIS cells, with 83% of the proliferating HMGA1-dense enhancers also being present in OIS. Hence, for simplicity, we referred to one set of HMGA1-dense enhancers (OIS) in the analysis, characterising them as having less H3K27ac than other enhancers (original Extended Data Fig. 2g, now Supplementary Fig. 3a). The gene enrichment of the proximal targets of these enhancers includes 'Epithelial-Mesenchymal Transition' genes (MSigDb Hallmarks), such as *LUM*, *FBN1* and *FGF2*, and also some cell cycle genes such as *CCNA2* and *PLK4*. Out of the 1,046 genes proximal to HMGA1-dense enhancers, 123 were differentially expressed in response to HMGA1 depletion by sh-HMGA1. We now added this information to the revised manuscript and the new Supplementary Fig. 3b.

In the original manuscript, we showed that the HMGA1-dense enhancers showed reduced overall connectivity compared to other enhancers during OIS (original Extended Data Fig. 5f, left, now Fig. 4j), consistent with their reduced H3K27ac content. We also mentioned that HMGA-dense enhancers showed no specific direction of interaction changes in OIS compared to proliferating cells (original Extended Data Fig. 5f, right, now Fig. 4k), reflecting the more general trend of A-A interactions showing both increases and decreases in OIS (original Fig. 2c, now Fig. 2e). We now followed up by determining gene promoters with increased or decreased interactions with the HMGA1-dense enhancers in OIS but found no clear trend of specific biological processes being affected, only that the genes with increased interactions with the enhancers were more AT-rich than the genes with decreased interactions, reflecting the more general classification of the regions involved in differential interactions.

Instead, HMGA1-dense enhancers showed overall increased interactions in OIS-shA1 cells compared to OIS (original Extended Data Fig. 5f, now Fig. 4k), reflecting the generally increased A-A interactions (original Fig. 2c, now Fig. 2e). Because of this, we focused on describing the overall effect of HMGA1 on A-A interactions, with 71% of the de novo increased interactions in OIS-shA1 involving enhancers or promoters, and the genes involved in these interactions (original Fig. 3g-i, now Fig. 4g-i), rather than classifying them by enhancer-promoter, promoter-promoter, and enhancer-enhancer interactions, or by their overlap with HMGA1-dense regions.

- We find out very little about the Hi-C data quality, read depth and resolution in the main text, which would be useful for assessing how cleanly HMGA1-relevant connections are portrayed in this data and at which resolution. The connections drawn are really interesting, but resolution and dynamic range of signal in the Hi-C data need to be well presented. More generally, I would have preferred more technical details from the Methods to make it out the main text to improve clarity.

We have now included in the main text and Supplementary Figures more information regarding the agreement between replicates and library size. We initially included this only in Methods because the Grow and OIS libraries were previously published and characterised in Olan et al. *Nat Comm.* 2020 PMID 33247104. The sh-A1 libraries were produced at the same time as the original samples. HMGA1 acts mainly via long-distance interactions, so we focused on matrices with low resolution (200kb bins). We now also included a figure to clarify the number of differential interactions and their overlap between the OIS - Grow and OIS-shA1 - OIS comparisons (new Fig. 2b).

At the beginning of the study, we investigated in detail a potential role for HMGA1 for more well-defined high-resolution interactions, similar to loop formation via CTCF/cohesin activity, or other structural proteins, due to its previously described role in the 'enhanceosome'. For this purpose, we also produced matched capture Hi-C libraries for the sh-A1 conditions at the same loci described in the previous study (Olan et al. *Nat Comm.* 2020 PMID 33247104), but we did not find any highly specific interactions between HMGA1 peaks, rather HMGA1 density seems to play a role in the 3D organisation. Although some differential contacts may be of interest in the capture Hi-C, such as *CCNA2* attaching to the nearby Lamin B1 domain in OIS and detaching in OIS-shA1, as well as the local dynamics at the *COL1A2* locus, we felt it was beyond the scope of the present study to include them.

- I find the trend lines in Fig. 2b to be not so convincing. It appears as if the change in curvature is mostly due to few interactions and not in the more dense part of the data cloud. In addition, I would not fully subscribe to the use of "homo-" and "heterotypic" interactions used here, as both regions more or less bind HMGA1, so are likely homotypic. Unless the authors can provide additional analysis (e.g., histone mark content) of these interacting regions, this panel should be either removed altogether or heavily revisited as regards its interpretation.

We have removed this panel from the figures and the description of interactions as homo-/heterotypical. However, we believe that the 'HMGA1 compatibility' is an important point linking HMGA1 binding density to the direction of the interaction changes. The link between HMGA1 levels, histone marks and interaction behaviour is described in more detail via the classification in the original Fig. 3b,c, now Fig. 3g-j.

- Not sure if "saddle plots" would make the message of Fig. 2C more apparent?

We have added the saddle plots as new Supplementary Fig. 3h, as well as retained the violin plots (now Fig. 2e), which make the local-distal distinction between the differential interactions, as HMGA1 is important for distal contacts. The log-fold changes may be more accurate for describing the inter-compartmental changes, as they were derived from a statistical model accounting for the biological replicates. Nevertheless, the saddle plots visually represent the general trend of dynamic inter-compartment interactions as seen in the violin plots. We thank the reviewer for this suggestion.

Reviewer Fig. R7 (new Supplementary Fig. 3h). Saddle plots of the average contact (distance-corrected) between regions binned by AB compartment score (negative values - B compartment, positive values - A compartment).

- The scRNA-seq data are a really nice addition, but their UMAP representation is now generally thought to be rather unrepresentative of data clustering. Would tSNE maps separate as nicely too? This is merely a suggestion. Moreover, I would like to see a statement in the main text on the apparent mixing of the OIS and OIS-shA1 populations, suggesting that HMGA1 regulation does not strongly change the senescent profile of the cells?

We only utilised the UMAP representation for visualisation. The clustering is determined using the Milo method, which only uses PCA projections to determine cell neighbourhoods and does not use the UMAP coordinates for the clustering of the over- and under-represented transcription profiles. We did not think t-sne plots showed a different relative positioning compared to the UMAP in terms of the distribution of cells across conditions (see below).

Reviewer Fig. R8. The t-sne projection of the scRNA-seq expression profiles.

We also believe this point is important and have been investigating alternative methods of measuring distances/similarities between gene expression profiles at single-cell level, as well as different projections, as the main point of contention raised against UMAP (Chari et al. *PLoS Comput Biol.* 2023 PMID [37590228](https://pubmed.ncbi.nlm.nih.gov/37590228/)) is the misrepresentation of actual distances/similarities between cells. The choice for a distance or similarity metric could similarly influence the comparisons between cells. However, we did not include this information as we believed this would be more suitable for a computational or methods study and requires a robust assessment across different datasets of the distance metrics used, which is beyond the current scope.

The reviewer is correct that “HMGA1 regulation does not strongly change the senescent profile of the cells”. This is consistent with the data (the current and previous studies) showing that HMGA1 reduction has little impact on senescence cell cycle arrest. We have clarified this point in the revised manuscript (see our response to reviewer #1-1).

- I would urge the authors to add the lung cancer data as a main figure given how well it rounds up their manuscript.

We thank the reviewer for the nice suggestion and we now included this in the main figure (new Fig. 6).

- Finally, I was wondering of HMGA1 overexpression in proliferating cells is doable and if this can impose some of the OIS effects to "young" cells? I know that HMGB1/2 overexpression is quite deleterious to IMR90 even in the short-term, but is this so for HMGA1 too? This would be a way to assess if HMGA1 levels alone are a driver of some of these changes? Currently, the only unclear point of this work lies with the fact that HMGA1 is present in both cell states, but in OIS is somehow redirected such that it can control a considerable part of the OIS program.

HMGA1 overexpression is possible using retro/lentivirus in normal fibroblasts, and the phenotype is drastically different depending on its level. When we use a strong promoter, such as CMV, ectopic HMGA1 is deleterious (Narita et al. *Cell* 2006 PMID 16901784). This supraphysiological level of HMGA1 can trigger a droplet-like structure and we recently showed the phase-separating activity of HMGA1 (Zhu et al. *Chembiochem*. 2023 PMID 36336658). To what extent HMGA1-mediated phase-separation contributes to SAHF formation in a physiological condition is an ongoing question of interest in our lab.

In contrast, a modest expression of ectopic HMGA1 using a weak promoter (e.g., wild-type LTR) shows little impact, at least in a short-term period - it slightly shortens the replicative lifespan in IMR90 cells (Narita et al. *Cell* 2006 PMID 16901784). We also previously showed that, while the modest HMGA1 overexpression is not sufficient for SAHF induction, it promotes SAHF-formation with simultaneous knockdown of lamin B1 (Sadaie et al. *Genes Dev.* 2013 PMID 23964094). Notably, this mild overexpression of HMGA1 is substantially higher than endogenous levels and yet the frequency of SAHF-positive cells in lamin B1-depleted IMR90 cells is lower than in OIS IMR90 cells (Sadaie et al. *Genes Dev.* 2013 PMID 23964094). Therefore, other factors are required to collectively induce the SAHF phenotype, likely contributing to the context-dependent role of HMGA1, as this reviewer points out.

This is a fundamental question and identifying the molecular basis of the shared (e.g., transcriptional buffer effects) and context-dependent (e.g., SAHF promotion) effects of HMGA1 will be a major direction of future research.

Reviewer #3:

General points

This is a well-executed study that digs into the biological roles of HMGA1 as a chromatin organizer, and how these interactions are modulated in senescence, and in turn, how this impacts the gene expression changes that are observed in senescence. The study exploits the elegant and well-characterized model of OIS, in which a stably integrated Ha-RasER protein is activated with tamoxifen. Superimposed on this is the downregulation of HMGA1, which is achieved by a transduction of a well-characterized shRNA against HMGA1 in a miR30 design. A diverse array of genomic methods are applied to analyze this biological system. The experiments are well executed and documented, and the analysis is comprehensive. The conclusions are largely consistent with prior information, but the data provide a much sharper focus and clarity. There are many interesting features that will be of interest to those in this field. The further elucidation of the chromatin architectural roles of HMGA1, especially how it shapes heterochromatin, will be wide interest. The one exception is that the data are not consistent with a role of HMGA1 as a direct regulator of gene expression. I view this also as a valuable contribution to the field.

I only have three specific points:

1) In a general sense, using only one shRNA sequence, controlled by empty vector (not scrambled sequence) is not rigorous enough to exclude off target effects. I understand that this specific shRNA has been the subject of several papers and that these issues have been addressed in the past. However, for completeness of this paper, this information should be summarized so that the reader can be reassured that off-target effects have been rigorously addressed. In my opinion a few sentences in the Methods section would suffice.

As the reviewer suggests, the sh-HMGA1 has been extensively characterized before, but we agree this is a valid point. Off-targets of sh-RNAs can be related to either target sequences (non-specific knockdown) or RNAi machinery (general toxicity, such as inflammatory reaction). At the time, we used a state-of-the-art program, which checks for sequences with more than three mismatches to any other genes (Paddison et al. *Nat Methods* 2004 PMID 16144086). We have since confirmed the specificity of the sh-RNA with more recent software.

For the second type of off-targets (sequence-unrelated), our shRNAs are not the original stem-loop shRNAs, which are more prone to this type of off-targets. We use a miR30 design, which incorporates the endogenous human miR-30 miRNA backbone (the 'empty' vector has the miR30 cassette without targeting oligos). The miR30 design has several advantages over the stem-loop shRNAs, including its integration of 'rule-based' approaches for selecting target sequences, e.g., destabilising the 5' end of the antisense strand, promoting strand-specific incorporation of miRNAs into the RNA-induced silencing complex (RISC) (Paddison et al. *Nat Methods* 2004 PMID 16144086). A more recent study identified a 'loop-counting rule' that ensures precise Dicer cleavage (Gu et al. *Cell* 2012 PMID 23141545). Implementing the endogenous 'rules' contributes to reduced off-targets.

Nevertheless, we cannot exclude potential off-target effects entirely. As suggested, we have added the following description in the Methods section: "The sh-HMGA1 has been extensively utilised, although its off-targets can not be completely excluded." We have also included the references describing the method for target sequence selection as well as reduced off-targets of the miR30 system compared to the stem-loop shRNAs (Paddison et al. *Nat Methods* 2004 PMID 16144086, Fellman et al. *Cell Rep.* 2013 PMID 24332856, Gu et al. *Cell* 2012 PMID 23141545). We thank the reviewer for this suggestion.

2) Given the wide-spread nature of HMGA1's effect on gene expression, some of the effects observed could be secondary (indirect) in nature (HMGA1 affecting the expression of some gene which in turn influences the particular result being measured). The good correlation of HMGA1 binding with many of the gene expression (and other) changes indicate that many of the effects are direct. However, indirect effects cannot be ruled, and I feel this should be briefly mentioned in the Discussion.

We have added the following sentence in Discussion accounting for the potential indirect effects of HMGA1 on transcription: "Despite the good correlation between 3D chromatin environments driven by HMGA1 binding density and gene expression changes, we do not exclude potential indirect effects of HMGA1 on transcription."

3) I found Fig. 4F confusing. Maybe the authors can try to redraw this? At minimum, I would like a more detailed explanation in the legend on how to interpret the figure.

We have redrawn this figure (see Reviewer Fig. R9 below and now Fig. 5f in the manuscript) to represent more clearly the main features of the four senescent clusters we identified and provided a more detailed legend.

Reviewer Fig. R9. (redrawn Fig. 4f, currently Fig. 5f in the revised manuscript): Schematic representation of the features of the four clusters of senescent cells identified using overrepresentation analysis (Fig. 4c, currently Fig. 5c), highlighting the clusters over-represented in OIS (2 and 4) and in OIS-shA1 (1 and 3), respectively. Notably, although inflammatory SASP (iSASP) and p16 have been collectively considered senescence hallmarks, they represent distinct types of senescence at the single-cell level. While clusters 3 and 4 express cell-cycle genes, cluster 3 is more prominent and thus may escape senescence. Cluster 4 resembles the previously described NOTCH-related 'early phase senescence' with augmented fibroblastic features (Hoare et al. 2016, Teo et al. 2019).

REVIEWERS' COMMENTS

Reviewer #1 (Remarks to the Author):

The authors have adequately addressed all my comments.

Reviewer #2 (Remarks to the Author):

The authors have done a thorough job in addressing all outstanding remarks. I would be happy to see the current version of the manuscript in print.

Reviewer #3 (Remarks to the Author):

I am satisfied with the responses to my issues that I raised in prior review. My recommendation is to accept the manuscript.